# Alphavirus-Driven Interferon Gamma (IFNg) Expression Inhibits Tumor Growth in Orthotopic 4T1 Breast Cancer Model

**DOI:** 10.3390/vaccines9111247

**Published:** 2021-10-27

**Authors:** Olga Trofimova, Ksenija Korotkaja, Dace Skrastina, Juris Jansons, Karina Spunde, Maria Isaguliants, Anna Zajakina

**Affiliations:** 1Latvian Biomedical Research and Study Centre, Ratsupites Str.1. k.1., LV-1067 Riga, Latvia; olj.trofimova@gmail.com (O.T.); ksenjakorotkaja@gmail.com (K.K.); dace.skrastina@biomed.lu.lv (D.S.); jansons@biomed.lu.lv (J.J.); spunde.carina@gmail.com (K.S.); 2Institute of Microbiology and Virology, Riga Stradins University, LV-1007 Riga, Latvia; maria.issagouliantis@rsu.lv; 3Department of Microbiology, Tumor and Cell Biology, Karolinska Institutet, 17177 Stockholm, Sweden; 4N.F. Gamaleya National Research Center for Epidemiology and Microbiology, 123 098 Moscow, Russia

**Keywords:** interferon gamma, cancer immunotherapy, viral vectors, alphavirus, bone-marrow-derived macrophages, spheroids, CD38, Pam3CSK4

## Abstract

Interferon gamma (IFNg) is a pleiotropic cytokine that can potentially reprogram the tumor microenvironment; however, the antitumor immunomodulatory properties of IFNg still need to be validated due to variable therapeutic outcomes in preclinical and clinical studies. We developed a replication-deficient Semliki Forest virus vector expressing IFNg (SFV/IFNg) and evaluated its immunomodulatory antitumor potential in vitro in a model of 3D spheroids and in vivo in an immunocompetent 4T1 mouse breast cancer model. We demonstrated that SFV-derived, IFN-g-stimulated bone marrow macrophages can be used to acquire the tumoricidal M1 phenotype in 3D nonattached conditions. Coculturing SFV/IFNg-infected 4T1 spheroids with BMDMs inhibited spheroid growth. In the orthotopic 4T1 mouse model, intratumoral administration of SFV/IFNg virus particles alone or in combination with the Pam3CSK4 TLR2/1 ligand led to significant inhibition of tumor growth compared to the administration of the control SFV/Luc virus particles. Analysis of the composition of intratumoral lymphoid cells isolated from tumors after SFV/IFNg treatment revealed increased CD4^+^ and CD8^+^ and decreased T-reg (CD4^+^/CD25^+^/FoxP3^+^) cell populations. Furthermore, a significant decrease in the populations of cells bearing myeloid cell markers CD11b, CD38, and CD206 was observed. In conclusion, the SFV/IFNg vector induces a therapeutic antitumor T-cell response and inhibits myeloid cell infiltration in treated tumors.

## 1. Introduction

The tumor microenvironment (TME) and immune cell composition play essential roles in tumor development. Immunotherapy is a novel strategy for cancer treatment aimed at modifying the TME and (re)programming immune cells. Pleotropic cytokines represent key instruments of immunotherapy because they allow the programming of the TME for cancer treatment due to their ability to mediate communication between cells and modify their functions [1,2]. IFN-gamma (IFNg) has crucial impacts on the organism immune response to tumors, including strong antiproliferative effects, modulation of adaptive and innate immune responses, activation of antigen-presenting cells by promoting the expression of MHC class I and II molecules on the cell surface, regulation of functions of T helper cells, activation of NK cells, and the ability to modify the functions of macrophages [3,4]. During the last 20 years, preclinical and clinical studies have examined the therapeutic impacts of IFNg alone or in combination with chemotherapy and immunotherapy. Despite these efforts, there is still no definite conclusion on the efficacy of IFNg in cancer therapy [5,6,7]. One of the drawbacks is systemic toxicity [6]. Furthermore, the IFNg-related induction of proinflammatory responses was shown to lead to not only antitumor but also protumor effects [8].

Vector-based intratumoral delivery of IFNg significantly increases therapeutic outcomes and reduces systemic toxicity. IFNg delivery by adenovirus [9,10], herpesvirus [11], and replication-deficient recombinant avian (fowlpox) virus vectors [12] showed promising results in animal models. Moreover, clinical trials testing the antitumor effects of adenoviral vectors expressing IFNg reported positive outcomes in T and B cell lymphoma patients [13,14]. Intratumoral-vector-based expression of IFNg not only reduced therapy toxicity but also restored the functions of immune cells in the TME [15,16].

Recent reports indicated that tumor progression largely depends on the (activities of) tumor-associated myeloid lineage cells [17]. Myeloid-derived cells, as a multifunctional and highly heterogeneous cell population, have become the subject of extensive studies within the last five years, which has led to reconsideration or refining of the classic concepts of immune crosstalk in tumors and revision of the role of IFNg and other cytokines in tumor development [8,18,19,20]. IFNg orchestrates leukocyte maturation and proinflammatory activation of myeloid cells. Classically, IFNg targets monocytes or macrophages and promotes their activation to the antitumoral M1 phenotype, whereas cancer cells promote myeloid cell infiltration and (re)program macrophages towards the tumor-supporting M2 phenotype [21]; however, several studies have indicated possible protumorigenic effects of IFNg through the induction of genomic instability (e.g., copy number alterations) or an immunoevasive gene expression signature in cancer cells (PD-L1, PD-L2, CTLA-4, nonclassical MHC class Ib antigens, IDO1, etc.), which correlated with clinical observations [22]. A promising way to enhance the antitumor effects of IFNg along with the inhibition of its potential protumor effects has been shown to be a complex treatment with immune checkpoint inhibitors [23,24,25] and chemoimmunotherapy [26,27] alone or in combination with oncolytic viral vectors, additionally inducing the innate immunity in tumors [28,29].

This study aimed to examine the antitumor effect of intratumoral expression of IFNg driven by alphaviral vector. Alphaviruses possess natural tumor tropism to mouse and human cancer cells, which has been documented in many preclinical studies [30,31]. Recent studies have indicated the synergistic antitumor activity of alphaviral vectors expressing cytokines when combined with checkpoint inhibitors (antibodies) and chemical drugs [32,33,34]. In contrast to other viruses, Semliki Forest virus (SFV) does not infect human and mouse macrophages [35], making the system suitable for functional SFV/IFNg-based programming of macrophages and the initiation of downstream M1-related proinflammatory reactions in the TME.

In previous studies, IFNg showed synergistic proinflammatory macrophage activation with viral RNA, bacterial endotoxin lipopolysaccharide (LPS), and different specific Toll-like receptor (TLR) agonists [36,37,38]. Recently, we demonstrated the ability of SFV-driven IFNg in combination with the TLR2/1 agonist Pam3CSK4 (a synthetic mimetic of bacterial lipopeptide—Pam3) to activate bone-marrow-derived macrophages (BMDMs), which inhibited Lewis lung carcinoma cell growth in cocultured monolayers [35]. We suppose that SFV-driven expression of IFNg will enhance the antitumoral IFNg effects through induction of innate antiviral immunity in tumors in response to SFV replication. In the current study, we characterized the ability of the SFV/IFNg vector to activate BMDMs towards the M1 phenotype and evaluated its effect on 4T1 mouse breast cancer cells in a three-dimensional (3D) spheroid model in vitro. Furthermore, we applied SFV/IFNg alone or in combination with Pam3 for in vivo therapy of 4T1 tumor-bearing mice to evaluate tumor growth inhibition. We explored the immune cell composition of the TME after SFV/IFNg treatment via flow cytometry and discovered the main changes occurring after treatment, which we believe to be responsible for the observed antitumor effects.

## 2. Materials and Methods

### 2.1. Cell Lines

For in vitro studies, 4T1-Fluc-Neo/eGFP-Puro cells (4T1/eGFP; Imanis Life Sciences, Rochester, USA) were cultured in RPMI-1640 selection medium (Cat. No. 12-115F; Lonza™ BioWhittaker™, Walkersville, MD, USA) supplemented with 10% fetal bovine serum (FBS; Cat. No. FBS-HI-12A; Capricorn Scientific, Ebsdorfergrund, Hessen, Germany), 2 mM L-glutamine (Cat. No. 25030-024, Gibco, Thermo Fisher Scientific, Waltham, MA, USA), 1% penicillin/streptomycin (pe/st) (Cat. No. 15070-063, Gibco), 0.1 mg/mL G418 (Cat. No. 10131-027, Gibco, Life Technologies, UK), and 2 µg/mL puromycin (Cat. No. ant-pr; InvivoGen, San Diego, CA, USA); or for 3D cultivation in RPMI-1640 medium supplemented with 10% FBS, 2 mM L-glutamine, and 1% pe/st. BHK-21 cells were cultured in BHK-Glasgow MEM (Cat. No. 21710-025; Gibco, Life Technologies, UK) supplemented with 5% FBS, 10% tryptose phosphate broth (Cat. No. 18050-039; Gibco, Life Technologies), 2 mM L-glutamine, 20 mM HEPES (Cat. No. 15630-056, Gibco, Life Technologies), and 1% pe/st.

L929 cells (ATCC^®^ CCL-1™, Manassas, VA, USA) producing murine macrophage colony-stimulating factor (M-CSF) were cultured to generate conditioned medium containing M-CSF. L929 cells were plated in RPMI-1640 with 10% FBS, 2 mM L-glutamine, and 1% pe/st in T175 flasks (Cat. No. 156502, Gibco) to achieve 60–70% confluence. Then, the medium was replaced with 30 mL of fresh medium and the cells were incubated for 8–9 days in a humidified 5% CO_2_ incubator at 37 °C. After cultivation, the conditioned medium (CM) from L929 cell culture was collected, centrifuged at 400× *g* for 10 min, transferred to a new tube, then centrifuged again at 10,000× *g* for 20 min at 4 °C to remove cell pellets. The CM preparations of L929 cell medium were aliquoted and stored at −20 °C.

In vivo studies were performed on murine mammary gland adenocarcinoma 4T1 cells (ATCC^®^ CRL-2539™) and 4T1 cells expressing firefly luciferase 4T1(Luc2) (Bioware Ultra Cell Line 4T1luc2, Caliper Life Sciences Inc., Hopkinton, MA, USA). Adenocarcinoma 4T1 cells were cultured in DMEM-GlutaMAX (Cat. No. 31966-021, Gibco), 10% FBS, and 40 µg/mL gentamicin (Cat. No. 00-0442; Sopharma, Sofia, Bulgaria), while 4T1(Luc2) cells were cultured in RPMI-1640 supplemented with 10% FBS, 2 mM L-glutamine, and 1% pe/st. Cells were incubated in a humidified 5% CO_2_ incubator at 37 °C and passaged using 0.05% trypsin solution (Cat. No. 15400-054, Gibco).

### 2.2. Formation of 4T1/eGFP Spheroids

Three-dimensional (3D) scaffold-free 4T1/eGFP cell culture spheroids were generated in 96-well black round bottom ultralow attachment plates (Cat. No. CLS4515, Corning, Life Sciences, New York, NY, USA) according to the manufacturer’s instructions. Briefly, 4T1/eGFP cells were collected from monolayers by trypsin treatment, filtered through 40 µm cell strainers (Cat. No. CLS431750, Corning, USA), then plated at 3000 cells per well in a total volume of 100 µL of RPMI-1640 medium supplemented with 10% FBS, 2 mM L-glutamine, and 1% pe/st. The cells were incubated in a humidified 5% CO_2_ incubator at 37 °C for 18–24 h, then the spheroids were used for infection with SFV vectors.

### 2.3. Expression Vectors

The pSFV/Luc plasmid (a kind gift of A. Merits, University of Tartu, Estonia) has been previously described [39]. The pSFV1 and pSFV-Helper1 plasmids [40] were generously provided by H. Garoff (Karolinska Institute, Stockholm, Sweden). The pSFV/IFNg and SFV/DS-Red vectors were generated in our lab as described previously [35,41].

### 2.4. Production of SFV Virus Particles

The plasmids pSFV/IFNg, pSFV/Luc, and pSFV-Helper1 were linearized with the restriction enzyme SpeI (Cat. No. ER1251; Thermo Fisher Scientific, Waltham, MA, USA). One microgram of each linearized plasmid was used for in vitro SFV-Helper1 RNA and recombinant RNA (SFV/IFNg, SFV/Luc, SFV/DS-Red) synthesis using SP6 RNA polymerase (Cat. No. AM2071; Thermo Fisher Scientific) as described previously [41]. For packaging of RNAs into viral particles, each recombinant RNA was co-electroporated (850 V, 25 μF, two pulses) with SFV-Helper1 RNA into 1 × 10^7^ BHK-21 cells using a Gene Pulser-II apparatus (Bio-Rad, Hercules, CA, USA). The electroporated BHK-21 cells were cultured in T75 tissue culture flasks in 15 mL of BHK-Glasgow MEM supplemented with 5% FBS medium in a humidified 5% CO_2_ incubator at 33 °C for 48 h. After cultivation, virus-containing supernatant was harvested from BHK-21 cells, clarified by centrifugation (Eppendorf fixed-angle rotor 5804, 10,000× *g*, 20 min, 4 °C), filtered through a vacuum 0.22 µm pore filter (Cat. No. 83.1822.001, Sarstedt, Newton, NC, USA), and concentrated by ultracentrifugation through double sucrose cushions (50% and 20% sucrose in TNE buffer) as previously described [42]. Briefly, 30 mL of filtered virus-containing medium was carefully overlayed onto a sucrose step gradient and centrifuged using a SW 32 Ti rotor (Beckman Coulter, Brea, CA, USA) at 150,000× *g* for 90 min at +4 °C. The virus-containing fractions (2 mL) were collected from the bottoms of the pierced tubes and dialyzed using dialysis cassettes (Cat. No. 02906-36; Spectra/Por, USA) for 4–5 h against TNE buffer. Virus preparations were aliquoted, frozen in liquid nitrogen, and stored at −80 °C. The viral titers (infectious units per mL, i.u./mL) were quantified by infecting BHK-21 cells seeded in 24-well plates. The day after infection, viral titres were quantified by immunostaining with anti-SFV nsp1 antibody (a kind gift from Prof. A. Merits, Tartu, Estonia) as previously described [39].

### 2.5. Isolation and Culturing of Bone-Marrow-Derived Macrophages (BMDMs)

Murine BMDMs were isolated from bone marrow progenitors obtained from BALB/c mice as previously described [43,44]. Briefly, mouse femurs and tibias were dissected from 8- to 10-week-old BALB/c mice and bone marrow cells were collected by flushing the femurs and tibias with 2–5 mL of RPMI-1640 supplemented with 10% FBS, 2 mM L-glutamine, and 1% pe/st (RPMI 10% FBS) using a 25 G needle. After the cells were centrifuged for 5 min at 400× *g*, the erythrocytes were lysed in 3 mL of lysis buffer (Cat. No. A10492-01; Gibco, Life Technologies) for 5 min at RT. The activity of lysis buffer was stopped with 10 mL of RPMI with 10% FBS and centrifuged at 400× *g* for 5 min. Next, the cells were resuspended in 10 mL of RPMI 10% FBS, then the cell suspension was filtered through a 70 µm cell strainer (Cat. No. 800070; BioSwisstec, Schaffhausen, Switzerland) and centrifuged at 400× *g* for 5 min. The cells were seeded onto 100 mm untreated cell culture dishes (Cat. No. 0030702018; Eppendorf, Hamburg, Germany) in complete BMDM cultivation medium (RPMI-1640 with 10% FBS and 30% L929-CM containing M-CSF, 2 mM L-glutamine, 1% pe/st) at a concentration of 8 × 10^5^ cells/mL and cultured for 7 days at 37 °C and 5% CO_2_. After 7 days of cultivation, cells that were not immediately used in the experiment were frozen and stored in liquid nitrogen.

Cryotubes containing BMDMs were removed from liquid nitrogen storage and immediately placed in a 37 °C water bath. After the cells were taken up from frozen stasis, they were immediately added to 12 mL of 37 °C prewarmed RPMI with 10% FBS solution, mixed by being inverted several times, then centrifuged for 7 min at 400× *g*. After centrifugation, the cells were washed with 10 mL of RPMI with 10% FBS twice and seeded in 100 mm untreated cell culture dishes in BMDM complete growth media containing 30% L929 CM. Cells were cultivated for 6 days at 37 °C and 5% CO_2_, t the medium was changed two times. On day 10, cells were detached from the plate with cold PBS, washed, and resuspended in complete media containing 10% L929 CM. Then, cells were counted and seeded in a monolayer (2D) onto 12-well plates (Cat. No. 0030721012; Eppendorf) at a concentration of 2 × 10^5^ cells/mL per well, or into a 96-well black round bottom ultralow attachment plate in the amount of 10^5^ cells per well in 200 µL, in order to provide 3D conditions for cell incubation at a concentration of 10^5^ per well in 200 µL solution. Cells were incubated overnight at +37 °C and 5% CO_2_ and then used for macrophage polarization experiments.

### 2.6. BMDM Polarization towards the M1 Phenotype with Virus-Derived IFNg (vdIFNg)

Cell culture supernatant containing vdIFNg was obtained by infection of BHK-21 cells with SFV/IFNg virus at a multiplicity of infection of 1 (MOI = 1) as previously described [35]. The control supernatant of infected cells (vdLuc-control, not containing IFNg) was obtained by infection of BHK-21 cells with SFV/Luc virus under the same conditions (MOI = 1). The vdIFNg in the supernatant was quantified by mouse IFNg ELISA (Cat. No. 88-7314-22; Invitrogen, Waltham, MA, USA). The supernatants were aliquoted, frozen, and subsequently used for 2D and 3D macrophage polarization experiments. According to the IFNg ELISA, the stock solution contained 2 µg/mL vdIFNg.

For M1 polarization, BMDM medium was supplemented with vdIFNg and Pam3 (Cat. No. tlrl-pms; InvivoGen, California, USA) to achieve final concentrations of 50 ng/mL vdIFNg and 50–100 ng/mL Pam3, respectively. In the control experiments, the BMDMs were cultured under 2D or 3D conditions in BMDM complete medium supplemented with an equivalent amount (µL) of supernatant collected from the SFV/Luc-infected BHK-21 cells (vdLuc). Untreated BMDMs (M0) served as an additional control. Cells were incubated for 2 days at +37 °C 5% CO_2_.

### 2.7. Analysis of BMDM Polarization by Flow Cytometry

After two days of macrophage incubation with vdIFNg and vdLuc (control), the cells were harvested for further flow cytometry analysis. For collection of cells from 96-well ultralow attachment plates (3D culture), the content of each well was mixed by pipetting and transferred into an 1.5 mL Eppendorf tube. The cells from at least 4 wells were combined and centrifuged at 500× *g* for 10 min to obtain a sufficient number of cells for immunostaining and flow cytometry. For detachment from 12-well plates (2D culture), wells were treated with cold PBS at 4 °C for 20 min. Then, the contents of the wells were flushed by pipetting, collected into an Eppendorf tube, and centrifuged at 500× *g* for 10 min. Cells from at least two wells were combined.

For immunostaining, the cells (2D and 3D) were washed twice with PBS containing 10% FBS (PBS-FBS), resuspended in 100 μL of PBS-FBS containing 12.5 μg/mL mouse IgG (Cat. No. I8765-5MG, Sigma-Aldrich, Co., LLC, St. Louis, MO, USA) to block nonspecific antibody binding, and incubated for 30 min at 4 °C. After blocking, the cells were washed with PBS-FBS and stained in 50 μL of PBS-FBS with the fluorophore-labeled monoclonal antibodies anti-CD11b-FITC (Cat. No. 11-0112-82; Invitrogen), anti-MHC II-PE (Cat. No. 12-5321-82; Invitrogen), anti-CD206-BV421 (Cat. No. 141717; Biolegend) and anti-CD38-PerCP-eFluor 710 (Cat. No. 46-0381-82; Invitrogen), which were diluted as recommended by the manufacturers. Cells were incubated with antibodies for 1 h at 4 °C, washed twice with PBS-FBS, then intracellular staining was performed with anti-iNOs-APC-eFluor 780 (Cat. No. 47-5920-82) and anti-Arginase 1-APC antibodies (Cat. No 17-3697-82; both from Invitrogen). The PerFix-nc kit (Cat. No. B31168; Beckman Coulter) was used according to the manufacturer’s instructions. Briefly, the cells were suspended in 25 μL of FBS, 15 μL of Fixative reagent was added, then the mixture was vortexed and incubated for 15 min at room temperature. Next, 150 μL of permeabilizing reagent was added to each tube, then immediately after membrane permeabilization the anti-iNOs and anti-Arginase 1 antibodies were added in recommended amounts and incubated for 30 min at room temperature in the dark. Finally, 1.8 mL of final reagent solution was added to the cells. Stained cells were kept at +4 °C and analyzed the next day using a FACSAria BD Hardware flow cytometer using FACSDiva Software (BD Biosciences, San Jose, CA, USA). The experiment was repeated twice and each staining was performed in duplicate. The data were analyzed by FlowJo 10.3 software (FlowJo LLC, Ashland, OR, USA) and presented as the mean of two independent experiments.

### 2.8. Nitric Oxide Assay

To determine the level of nitric oxide in activated BMDM cell culture medium, we used a nitric oxide assay kit (Cat. No. EMSNO; Invitrogen). The determination of the amount of nitric oxide was based on the detection of nitrite levels in cell culture media. Briefly, 50 µL of cell culture media was collected from each well, clarified by centrifugation, then used for NO quantification. The nitrite standards provided by the kit were used for standard curve generation and NO quantification. The optical density was measured at 540 nm using a spectrophotometer. The data are presented as the mean values of two independent experiments, with each sample tested in triplicate.

### 2.9. Infection of 4T1/eGFP Spheroids with Recombinant SFV Viruses

The SFV/DS-Red, SFV/Luc and SFV/IFNg virus stock solutions were diluted in PBS (containing Mg^2+^ and Ca^2+^, PBS-Ca/Mg) to achieve concentrations of 5 × 10^5^ (SFV/Luc and SFV/IFNg) and 1 × 10^6^ (SFV/DS-Red) viral particles (i.u.) per 1 mL. Spheroids were washed twice with PBS-Ca/Mg; 200 μL of PBS-Ca/Mg was added to each well to the spheroids grown in 100 µL of the cultivation medium and carefully removed immediately after avoiding the loss of the free-floating spheroids. Next, 100 μL of the solution containing virus particles was added to each well to achieve 5 × 10^4^ (SFV/Luc and SFV/IFNg) or 1 × 10^5^ (SFV/DS-Red) i.u./well. The control cells were incubated with PBS-Ca/Mg. The spheroids were incubated at 37 °C for 1 h and 10 min on a shaker at 40 rpm (3D Sunflower mini-shaker, BS-010151-AAG, Bio-San, Riga, Latvia); additionally, infection without shaking was tested. After incubation, 100 μL of virus-containing solution was removed from each well and 150 μL of RPMI with 10% FBS was added. Immediately after, 150 μL of cell media was removed from each well and replaced with fresh 100 μL of RPMI supplemented with 10% FBS. Infected spheroids were incubated in a humidified 5% CO_2_ incubator at 37 °C.

### 2.10. Spheroid Confocal Microscopy

For confocal fluorescence microscopy, 4T1/eGFP spheroids were infected with SFV/DS-Red virus as described above, cultured for 2 days at 37 °C and 5% CO_2_, and subjected to confocal laser scanning microscopy using a Leica TCS SP8 Laser DPSS561 with a 70.7 μm pinhole and scan speed of 400 Hz. The fluorescence was detected as follows: eGFP excitation laser 488 nm and emission detector PMT 2 (493–560 nm); Ds-Red excitation laser 561 nm and emission detector HyD (573–651 nm). The fluorescence intensity profiles from the spheroid upper rim to the bottom were acquired using a z-stack of 48 focal planes with a step pass of 5 μm. The images were analyzed by LasX 3.1.5 software. Total fluorescence intensity (eGFP and DS-Red profiles) was calculated for each spheroid and at least four spheroids were analyzed in each group. Through imaging, all measurement conditions were kept constant for all experiments. The experiment was repeated twice.

### 2.11. Fluorimetry of 4T1/eGFP Spheroids Infected with SFV Vectors and Cocultured with BMDMs

The 4T1/eGFP spheroids (3000 cells/well) were infected with SFV/IFNg or SFV/Luc virus at a virus dose of 5 × 10^4^ i.u./well without plate shaking as described above. The next day after infection, the spheroid supernatants (50 µL) were collected to measure the IFNg production by ELISA (Cat. No. 88-7314-22; Invitrogen). Then, the BMDMs were added to the spheroids at a concentration of 3 × 10^4^ cells in 100 μL of BM medium (10% L929 CM) per well (day 0). Pam3 ligand was added to wells to a final concentration of 50 ng/mL. The dynamics of spheroid growth were measured by an eGFP fluorimetry assay using Victor3V 1420-040 Multilabel HTS Counter (PerkinElmer, Waltham, MA, USA) with an emission filter of 485 nm and a detection filter of 535 nm. To prevent liquid evaporation from wells, we added 40–50 µL of fresh spheroid medium to each well every second day after fluorimetry. The total fluorescence was measured every second day from individual measurements (each group, *n* = 6–8); the data were presented as the mean fluorescence (a.u.) ± standard deviation. The experiment was repeated twice. Additionally, for microscopy visualization, detached macrophages were labeled with fluorescence dye (CellTracker CM-DiI, Cat. No. C7001, Thermo Fisher, or a similar dye providing unspecific labeling of cells), washed with cell medium, added to the infected spheroids as described above, and cultivated for at least 7 days until macrophage labeling was detectable. For visual control of spheroid growth, fluorescence microscopy of the spheroids was performed with labeled or unlabeled macrophages using a Leica DM-IL inverted contrasting microscope (Leica Microsysystems, Wetzlar, Germany). All experiments with labeled macrophages were repeated at least three times.

### 2.12. Experiments with Animals and In Vivo Imaging

Female BALB/c mice (6–7 weeks of age) were purchased from the Laboratory Animal Center, University of Tartu (Tartu, Estonia). The mice were housed 5 per cage in a climate-controlled room (temperature 22 ± 2 °C and humidity 50 ± 10%) under a 12 h light/dark cycle and provided a standard diet and water ad libitum. All animal experimental protocols were approved by the Latvian Animal Protection Ethical Committee of Food and Veterinary Service (Permit Nr. 93, from 11 December 2017, Riga, Latvia).

### 2.13. Coinjection of 4T1(Luc2) Cells with BMDMs

Treatment of 4T1 tumors with M1 macrophages polarized by vdIFNg was tested in the orthotopic (orth.) murine model. For this, before the implantation, aliquots of 1 × 10^4^ cells 4T1(Luc2) cells were mixed with either 2 × 10^4^ M0 (4T1(Luc2)+M0) or 2 × 10^4^ M1-like cells activated with vdIFNg/Pam3 (4T1(Luc2)+M1) in a total volume of 50 µL. BALB/c mice (*n* = 5 per group) were injected with 50 ul of either 4T1(Luc2)+M0 or 4T1(Luc2)+M1 cell suspensions into the right thoracic mammary gland fat pads (day 0).

For bioluminescence imaging, mice received an intraperitoneal injection of 200 μL of D-luciferin potassium salt solution (XenoLight D-Luciferin, PerkinElmer) in PBS at a dose of 0.15 mg of D-luciferin per 1 g weight of each animal. Ten minutes after the injection of D-luciferin, the mice were anesthetized with isoflurane/oxygen and placed in an in vivo imaging system (IVIS Spectrum, Perkin Elmer) as described by us previously [45]. The bioluminescence from injection sites was assessed every second day. After 15 days, the mice were anesthetized and humanely sacrificed, then the tumors were dissected and weighed. The lungs were removed and placed in 24-well plates to monitor infiltration of tumor cells (IVIS, Perkin Elmer) as previously described [46]. The data were analyzed using the total photon flux emission (photons/second) in the regions of interest (ROI) using Living Image^®^ version 4.5 (PerkinElmer).

### 2.14. Treatment of 4T1 Tumors with SFV Vectors

Treatment of 4T1 tumors with SFV vectors was tested in both orthotopic and subcutaneous (s.c.) 4T1 models of breast cancer. In the orth. model, 4T1 cells were similarly suspended in PBS, and 50 µL of cell suspension containing 1.25 × 10^5^ cells/50 µL was injected into the right thoracic mammary gland fat pads (day 0). In the sc. model, 4T1 cells were suspended in PBS and 100 µL of cell suspension containing 2.50 × 10^5^ cells/100 µL was subcutaneously injected above the right shoulder blade of each mouse (day 0). When the tumors became palpable—day 7 for s.c. tumors and day 4 for orth. tumors—an intratumoral (i.t.) injection (100 µL) of SFV vectors (or PBS as a control) was performed with 4 × 10^7^ i.u. of SFV/IFNg or SFV/Luc per tumor, respectively. The repeated vector i.t. injections were performed at day 13 (s.c. tumors) and at day 10 (orth. tumors) with the same virus dose. The next day after virus treatment mice were intratumorally injected with Pam3 ligand (Pam3CSK4) dissolved in PBS at 10 µg/60 µL and 15 µg/90 µL per tumor. The tumor diameters were measured using digital electronic calipers and the tumor volume in mm^3^ was calculated using the following formula: V = (width^2^ × length)/2. At day 17 (s.c. tumors) and at day 14 (orth. tumors), the animals were anesthetized and sacrificed, then the tumors were removed, weighed, and subjected to immune cell isolation for flow cytometry. The tumor inhibition rate (IR, %) was calculated as follows: IR = 100−(mean weight of treated tumors/(mean weight of PBS control tumors × 100%).

### 2.15. Flow Cytometry Analysis of Intratumoral Immune Cells

The general protocol used for flow cytometry analysis of 4T1 tumors has been described elsewhere [47,48,49]. Tumors were homogenized with 3–4 mL of collagenase A (Cat. No. 10103586001; Roche, Basel, Switzerland) at 1.5 mg/mL and DNase at 15 µg/mL (Cat. No. ENZ-417; ProSpec Medical Holding, Los Angeles, CA, USA) in DMEM and incubated for 1 h on a magnet stirrer at 37 °C. After the enzymatic reaction was stopped with 8–10 mL of ice-cold DMEM with 10% FBS, the cells were filtered through a 70 μm strainer and centrifuged at 400× *g* for 10 min. Then, the cells were resuspended in 3 mL of erythrocyte lysis buffer (Cat. No. A10492-01; Gibco, Life Technologies) and incubated for 5 min at RT. The activity of the lysis buffer was stopped with 10 mL of RPMI with 10% FBS. Next, the cells were centrifuged at 400× *g* for 5 min and washed with 7–8 mL PBS-FBS twice. Then, the cells were counted with a Countess Automated Cell Counter (Thermo Fisher Scientific Invitrogen) and 1 × 10^6^ cells per staining were used for further procedures. The cells were resuspended in 100 µL of blocking solution containing 12.5 μg/mL mouse IgG diluted in PBS-FBS and incubated for 30 min on ice. After blocking, the cells were washed with PBS-FBS and stained with fluorochrome-labeled monoclonal antibodies diluted as recommended by the manufacturers in 50 μL of PBS-FBS per staining. The following monoclonal antibodies were used: for mix 1—anti-CD11b-FITC, anti-MHC II-PE, anti-CD206-BV421, and anti-CD38-PerCP-eFluor 710; for mix 2—anti-CD3-FITC (Cat. No. 11-0032-82), anti-CD4-APC-eFluor 780 (Cat. No. 47-0041-82), anti-CD8-APC (Cat. No. 17-0081-82), and anti- CD25-PE (Cat. No. 12-0251-82) (all from Invitrogen). The cells with antibodies were then incubated for 1 h at +4 °C. After staining, the cells were washed twice with PBS-FBS.

For intracellular staining with anti-iNOs-APC-eFluor 780, anti-Arginase1-APC, and anti-FoxP3-PerCP-Cyanine5.5 (Cat. No. 45-5773-82; Invitrogen) antibodies, the PercFix-nc kit was used according to the provided instructions. Briefly, the cells were suspended in 25 μL of FBS, then 15 μL of Fixative reagent was added, vortexed, and incubated for 15 min at room temperature. Then, 150 μL of permeabilizing reagent was added to each tube. Immediately after membrane permeabilization, anti-iNOs and anti-Arginase 1 were added to mix 1 and anti-FoxP3 was added to mix 2 at an appropriate dilution and incubated for 30 min at room temperature in the dark. Finally, 1.8 mL of final reagent solution was added to the cell suspension. Stained and fixed cells were stored at +4 °C and analyzed within two days by FACSAria BD Hardware and BD FACSDiva Software. UltraComp eBeads™ (Cat. No. 01-2222; Invitrogen, Thermo Fisher Scientific) were used for the compensation matrix. Importantly, the tumor homogenization, cell isolation, and respective staining with antibodies were performed simultaneously within one day for all compared groups.

### 2.16. Flow Cytometry Data Analysis

Data were analyzed with FlowJo software version 10.3 (FlowJo, Ashland, OR, USA). The total population was determined based on SSC-A and FSC-A. Single cells were determined by FSC-A and FSC-H to exclude cell aggregates. The gate of each antibody was determined using an unstained control, after which the percentage of the expression level of each marker was defined.

### 2.17. Statistical Analysis

The statistical analysis was performed using GraphPad Prism 7 software. Confocal images were statistically compared by two-way ANOVA and Tukey’s multiple comparisons test. In vitro fluorimetry data were analyzed by *t*-tests and two-way ANOVA. In vivo tumor growth and flow cytometric data were analyzed by the Mann–Whitney nonparametric *t*-test; bioluminescence signals were analyzed using repeated-measures two-way ANOVA Sidak’s multiple comparisons test. Here, *p*-values of 0.05 or less were considered statistically significant.

## 3. Results

### 3.1. Generation of Cancer Cell Spheroids and Their Infection with SFV Vector (SFV/DS-Red)

Three-dimensional spheroids made from cancer cells are a relevant system for investigating the interactions between cancer cells, macrophages, and the SFV/IFNg vector in this study. The general plan of the in vitro research was to establish a 3D model to evaluate whether the SFV/IFNg vector, through infection of 4T1 mouse breast cancer cells, can induce a tumor suppressive phenotype in macrophages (M1), followed by the assessment of the therapeutic potential of the SFV/IFNg vector in vivo.

First, we evaluated the ability of the SFV vector to deliver transgenes to spheroids under 3D infection conditions. For this purpose, we generated eGFP-producing 4T1 spheroids (4T1/eGFP) cultured for 18 h in nonadherent 96-well plates at a concentration of 3 × 10^3^ cells in 100 µL. The cancer cells, in contrast to macrophages, aggregated and formed tight 120–150-µm-diameter single spheroids within 18 h of incubation of the cell suspension in each well. The 4T1/eGFP spheroids were infected with SFV/DS-Red virus (1 × 10^5^ i.u. per well) and DS-Red gene expression was analyzed by live fluorescence confocal microscopy at 48 h post-infection (Figure 1a, maximum intensity projection images). The confocal z slices of each spheroid were acquired every 5 µm for a total of 48 imaging planes and the mean fluorescence intensity of every confocal slice was plotted as a function of its z depth (Figure 1b). The image analysis revealed that the DS-Red-positive cells were mostly located on the surface of the spheroids with nonhomogeneous penetration into the spheroid (Figure 1b). As expected, shaking of the spheroid plate during incubation with a virus significantly enhanced the infection efficiency (total DS-Red fluorescence intensity) and penetration of the virus into deeper spheroid slices (Figure 1b,c). We concluded that the SFV vector infects the 4T1 spheroids in 3D conditions; however, virus spread within the spheroid was limited. Furthermore, confocal microscopy images revealed different growth patterns of infected and uninfected spheroids. Infected spheroids displayed less total eGFP fluorescence signals (*p* < 0.0001; Figure 2c) and the size of the spheroids was visually smaller, indicating the inhibitory effect of the infection.

### 3.2. SFV-Derived IFNg Activates BMDMs towards an M1-Like Phenotype in 3D Conditions

Next, we examined the ability of SFV virus-derived IFNg (vdIFNg, a supernatant from the cells infected with SFV/IFNg) to polarize macrophages to an M1-like tumor-suppressive phenotype and the ability of these M1 macrophages to inhibit 4T1 spheroid growth in a 3D model. Recombinant IFNg in the presence of TLR ligands (such as LPS, Pam3) was shown to polarize BMDMs to the M1 phenotype when the cells were seeded on a standard 2D attachment plate [38]. Here, for the first time, we evaluated M1 polarization under free-floating conditions (3D) compared to standard 2D cell cultivation using vdIFNg and Pam3 TLR ligands.

BMDMs were seeded either in 12-well attachment plates to achieve a monolayer (2D) or into 96-well ultralow attachment plates at a relatively high cell density (10^5^ cells per 96-well). The next day, the cell medium was supplemented with vdIFNg and Pam3 to trigger M1 polarization. Cell culture supernatant from SFV-infected cells may contain multiple mediators of the innate cell immune response (e.g., type I IFNs), which can potentially affect macrophage polarization. To account for their effect, we used control supernatants of BHK-21 cells infected with SFV/Luc virus (vdLuc; see methods). BMDMs treated with vdLuc supernatant were considered as M0 undifferentiated cells (“M0 control”). Practically, we did not detect any differences between vdLuc control and untreated M0 (PBS) macrophages in this study. Nevertheless, these controls were used in all experiments.

Polarized macrophages were subjected to nitric oxide assays and flow cytometry analysis. The ability to produce nitric oxide (NO) is one of the main characteristics of proinflammatory M1-like macrophages. To assess NO levels in cell culture media, nitrites, as the product of nitric oxide oxidation, were quantified by the Griess test. Nitric oxide assays showed that treatment of BMDMs with vdIFNg and Pam3 in 2D and 3D conditions induced strong NO production (Figure 2a). The maximal NO production, dependent on the number of seeded cells, was observed on days 2–3 of cultivation. Treatment with vdLuc supernatant (M0 control) did not enhance the production of nitric oxide by macrophages (Figure 2a).

Expression of macrophage polarization markers was analyzed by flow cytometry. The panmacrophage CD11b surface marker was detected in 80–90% of BMDMs and was decreased upon cultivation in 3D conditions (*p* < 0.05; Figure 2b). BMDMs activated by vdIFNg revealed a significant increase in the M1-like phenotype-specific markers MHCII, CD38, and intracellular inducible NO synthase (iNOs) (Figure 2b,c). While MHCII expression did not show significant differences under 2D and 3D conditions, the levels of the CD38 and iNOs M1 markers were higher in 3D than in 2D conditions (CD38, *p* = 0.0180 and iNOs, *p* = 0.0028; Figure 2b).

As expected, the level of CD206 (a classic marker of M2 macrophages [50]) did not show significant changes depending on cell activation status and was relatively low in all groups (<2%). Interestingly, the level of Arginase 1, which is considered a predominant marker of M2-like macrophages [51], was increased in both vdIFNg groups (2D and 3D, Figure 2b). Several studies have shown that overexpression of Arginase 1 in the M1 population could be due to the presence of the TLR2/1 ligand Pam3 [52].

In summary, we concluded that macrophages can be effectively activated to an M1-like phenotype under 3D free-floating conditions using vdIFNg/Pam3 treatment, which in general resembles the 2D characteristics of M1 macrophages.

### 3.3. SFV/IFNg Infection of the 4T1/eGFP Spheroids Inhibits Spheroid groWth in the Presence and Absence of Macrophages

The 4T1/eGFP spheroids were infected with equal amounts of either SFV/IFNg or SFV/Luc viruses or incubated with PBS. We used a relatively low virus dose for infection (5 × 10^4^ i.u./spheroid) to avoid significant inhibition of spheroid growth by virus infection itself, which may mask the inhibitory effects of macrophages. Moreover, spheroid infection with such an SFV/IFNg virus dose provided up to 15 ng/mL of vdIFNg production, as confirmed by quantitative anti-IFNg ELISA (Figure 3a), which is a sufficient amount for macrophage activation [35,38]. The day after spheroid infection with SFV, BMDMs (M0) were added (day 0). To confirm macrophage activation to the M1 pro-inflammatory phenotype, we assessed the presence of NO in cell media after two days of macrophage incubation with infected spheroids and the uninfected control (day 2), which showed NO production levels similar to those in 2D and 3D conditions without spheroids (Figure 2a and Figure 3a).

Spheroid growth was measured by fluorimetry every second day for 10 days to assess the inhibitory effects of macrophages in the presence or absence of SFV vectors (Figure 3b). For this, twelve combination groups of spheroids (sph) with or without macrophages (M0), SFV vectors, and Pam3 ligand were generated and the respective fluorimetry data were collected in six fluorimetry measurements starting from day 0 (Appendix A). Analysis of the fluorimetry data at day 10 showed that the strongest inhibitory effect on the cancer cell spheroid growth was observed in the sph+SFV/IFNg+Pam3+M0 and sph+SFV/IFNg+Pam3 groups (Figure 3b). Interestingly, macrophages and SFV/IFNg independently inhibited spheroid growth in all groups compared to the untreated or SFV/Luc-treated groups (sph and sph+M0 (*p* < 0.0001); sph+Pam3 and sph+Pam3+M0 (*p* < 0.0001); sph+SFV/Luc and sph+SFV/IFNg (*p* < 0.0001); sph+SFV/Luc+Pam3 and sph+SFV/IFNg+Pam3 (*p* < 0.0001); Figure 3b).

Very high (*p* < 0.0001) differences were revealed between untreated and treated spheroids, namely sph and sph+SFV/IFNg+Pam3+M0 and sph and sph+SFV/IFNg+Pam3 (Figure 3b). Although both SFV/IFNg and M0 showed inhibitory effects on the spheroid growth, on day 10 the inhibitory effect of the sph+SFV/IFNg+Pam3+M0 combination was significantly higher than that of the combination without macrophages (sph+SFV/IFNg+Pam3, *p* < 0.001), pointing to the inhibitory role of macrophages in the tested system. Measurement of NO in all groups at day 4 demonstrated that the sph+SFV/IFNg+Pam3+M0 combination triggered potent production of NO (40 µM; Appendix A). Furthermore, these data (Figure 3b, Appendix A) were supported by the quantification of luminescence signals produced by the spheroid lysates prepared at day 10 (Appendix A). Altogether, this demonstrated the inhibition of spheroid growth in the presence of SFV/IFNg and macrophages compared to the untreated controls (*p* < 0.0001).

We concluded that SFV infection in combination with macrophages provided the most significant inhibition of spheroid growth (Figure 3b). The inhibitory effect of SFV/IFNg was explained by the known cytostatic effect of IFNg, while the inhibitory effects of M0 on their own were unclear. To dissect the effects, we visualized 4T1/eGFP spheroid growth in the presence of fluorescently labeled macrophages (Figure 3c). The infection of spheroids with SFV/IFNg prevented the migration or distribution of 4T1/eGFP cells within the well. On the contrary, M0 macrophages stimulated migration of 4T1/eGFP cells out of the spheroids (Figure 3c, day 7; Appendix A, day 6 and day 10). This migration would eventually lead to diminished fluorescence signals from the spheroids, interpretable as inhibition of spheroid growth. The effect of Pam3 on spheroid growth remains to be investigated in a separate study.

### 3.4. Inhibition of Growth of 4T1luc2 Tumor Cells by M1 Macrophages in BALB/c Mice

Both M0 and SFV/IFNg vector macrophages demonstrated inhibitory effects on the spheroid growth, so it was unclear whether vdIFNg-activated M1 macrophages can preferentially suppress tumor growth compared to M0 macrophages. We sought to prove this in in vivo settings by evaluating the effect of M1-like macrophages prepolarized by vdIFNg treatment on the growth of 4T1 tumors in BALB/c mice.

We used an in vivo imaging system (IVIS) to measure tumor growth in mice in the presence of M0- or vdIFNg-polarized M1 macrophages, which were coinjected orthotopically together with 4T1(Luc2) cells (day 0). Tumor growth was monitored by in vivo bioluminescence imaging (BLI) to assess the photon flux emitted by 4T1(Luc2) cells every second day (Figure 4). Coinjection of 4T1luc2 cells with M1 macrophages significantly delayed tumor growth, as could be seen from the comparison of bioluminescent signals released from the sites of implantation of 4T1(Luc2)+M1 compared to 4T1(Luc2)+M0 cells (days 7 and 9 post-implantation; *p* < 0.05; Figure 4a,b). No difference in bioluminescence from 4T1(Luc2)+M1- compared to 4T1(Luc2)+M0-derived tumors was observed at the later time points when the bioluminescence signal was close to saturation (*p* > 0.1; Figure 4a,b); however, on day 15 when mice were sacrificed, the tumor weight in the M1 co-injected group was found to be significantly lower than in M0 group (*p* = 0.0278; Figure 4c).

We also assessed the infiltration of tumor cells into the distal organs, namely the lungs as the organs mostly affected in the 4T1/4T1luc2 model [45]. For this, we assessed the bioluminescent signals emitted by organs by ex vivo imaging performed immediately after their dissection as previously described [46]. The bioluminescent signals emitted by the lungs of 4T1(Luc2)+M1- and 4T1(Luc2)+M0-implanted mice did not differ (Figure 4b,c), indicating that M1 macrophages had no long-term effect on the migration of tumor cells registered by the experimental end-point.

To characterize the populations of the intratumoral immune cells, we homogenized tumors and subjected cell suspensions to the analysis by flow cytometry to identify the percentage of myeloid and lymphoid cells (Figure 5). Myeloid cells were characterized by the pan-myeloid marker anti-CD11b. We revealed a significant decrease in the CD11b^+^ cell population in 4T1luc2+M1 compared to 4T1luc2+M0 tumors (*p* = 0.0278; Figure 5a). Most of the CD11b^+^ cells were found to express MHCII (up-to 80%, Appendix A). M1 and M0 tumors did not differ in % of MHCII^high^ cells, neither within CD11b^+^ population, nor within total tumor cell population (*p* > 0.1; Appendix A). This was in contrast to the results of in vitro BMDM analysis, where MHCII receptor was highly expressed only by M1-polarized CD11b^+^ cells (Figure 2c). Interestingly, although M1 co-injected tumors demonstrated a decrease in the population of CD11b^+^ cells, the presence of CD11b^+^/CD38^+^ cells in M1 tumors was significantly higher than that in the M0 group (*p* = 0.0078; Figure 5b).

Cancer cells and tumor-associated fibroblasts reprogram macrophages to the tumor-promoting M2 phenotype [51]. Surprisingly, although CD206^+^/CD11b^+^ double positive cells did not show the statistically significant changes in total tumor cells of M0 or M1 coinjected tumors (Figure 5b), we observed an increase in CD206^+^ cells in CD11b^+^ populations in M1-coinjected tumors, *p* = 0.0159 (Appendix A). Furthermore, both arginase 1 (Arg 1^+^) and inducible NO synthase (iNOs^+^) intracellular markers increased in CD11b^+^ population of cells isolated from M1 coinjected tumors, *p* = 0.004; *p* = 0.0159, respectively, whereas in total cells only iNOs^+^/CD11b^+^ double positive cell population was increased, *p* = 0.0317, indicating on the pro-inflammatory profile of the myeloid cells (Appendix A).

The most important changes were revealed in the lymphoid cell population, specifically T-regulatory cells (T-regs). Although 4T1luc2+M1 and 4T1luc2+M0 tumors did not differ in populations of CD4^+^ and CD8^+^ T cells (Figure 5a), 4T1luc2+M1 tumors had significantly lower numbers CD25^+^/FoxP3^+^/CD4^+^ cells, indicating that M1 pretreatment diminishes the population of intratumoral T-regs (compared to treatment with M0, *p* = 0.0079; Figure 5a).

Thus, comparison of the M0 and M1 co-injected tumors clearly demonstrated that treatment with M1 inhibits tumor growth and changes composition of the myeloid and lymphoid intratumoral cell subsets by inhibiting infiltration of CD11b+ cells, increasing proinflammatory iNOs^+^ phenotype of the myeloid cells, and decreasing the number of T-regs in the TME.

### 3.5. Intratumoral Injection of SFV/IFNg Virus Inhibits Orthotopic 4T1 Tumor Growth

To evaluate the antitumoral potential of SFV/IFNg virus, we used orthotopic (orth.) and subcutaneous (sc.) 4T1 murine models (*n* = 5). Two intratumoral (i.t.) injections of the vector (4 × 10^7^ i.u./tumor) were performed: the first injection was given as soon as the tumors became palpable (day 7 for s.c. model; day 4 for orth. model); the second repeated injection was performed six days later (Figure 6a,b; red arrows indicate the vector/PBS injections). Furthermore, to stimulate macrophage polarization to M1, we treated the mice with Pam3 ligand. The day after virus vector administration, the mice received i.t. injections of the Pam3 ligand solution at 10 µg (first injection) and 15 µg (repeated injection) per tumor). Tumor growth was measured regularly at 17 (s.c.) and 14 (orth.) days. At the end of the experiments, the tumors were resected and weighed. The tumor growth curves for all individual animals are presented in Appendix A. Although in the s.c. model, a trend of tumor growth inhibition was observed in the SFV/IFNg+Pam3-treated group, the tumor growth parameters and the final tumor weights varied within each group; therefore, the observed inhibition did not reach the level of significance (Figure 6a). The statistical significance (Mann–Whitney *t*-test) of the tumor volume of groups treated with SFV/Luc (sc.SFV/Luc+Pam3) and SFV/IFNg (s.c.SFV/IFNg+Pam3) on the last day (day 17) reached the probability level of *p* = 0.2103 (nonsignificant).

In contrast to sc. tumors, the orthotopic model showed relatively homogeneous tumor growth parameters, demonstrating significant inhibition of tumor growth in the treated mice (Figure 6b). Compared to the PBS group, the inhibition of tumor growth in the mice treated with SFV/IFNg+Pam3 was stronger than that in the SFV/Luc+Pam3 group: PBS vs. SFV/IFNg+Pam3 with last day tumor volume of *p* = 0.004 and tumor weight of *p* = 0.004; PBS vs. SFV/Luc+Pam3 with last day 14 tumor volume of *p* = 0.0476 and the tumor weight of *p* = 0.0159. There was also a significant difference between the SFV/IFNg+Pam3 and SFV/Luc+Pam3 groups (last day tumor volume *p* = 0.004 and tumor weight *p* = 0.004). The tumor inhibition rate of the SFV/IFNg+Pam3 treatment group versus the PBS group was 59.6%, whereas SFV/Luc+Pam3 treatment reached only 27.8% versus the PBS group, indicating the therapeutic activity of IFNg.

To exclude the potential inhibitory effect of Pam3 on its own activity, we tested Pam3 administration without viral vectors (Figure 6c). Some studies revealed the antitumoral effect of Pam3 and other TLR agonists [53,54], but we did not observe any significant inhibition of 4T1 orthotopic tumor growth in the Pam3-treated mice in doses comparable or higher than those used in previous studies.

Since the SFV/Luc vector in combination with Pam3 showed a remarkable tumor-inhibitory effect, we also evaluated the antitumor potential of SFV vectors on their own without Pam3 administration. Our data demonstrated a significant inhibitory effect of SFV/IFNg without Pam3 compared to that of the PBS group (last day tumor volume *p* = 0.004 and the tumor weight *p* = 0.0079) and to the SFV/Luc group (last day tumor volume *p* = 0.0159 and the tumor weight *p* = 0.0476) (Figure 6d). There were no differences between the PBS- and SFV/Luc-treated groups in the absence of Pam3 (last day tumor volume *p* = 0.3452 and the tumor weight *p* = 0.5).

Remarkably, the tumor inhibition rate of the SFV/IFNg+Pam3 treatment group was higher than that of the SFV/IFNg alone group (59.6% and 49.1%, respectively, versus the corresponding PBS groups). SFV/Luc+Pam3 exhibited antitumor activity with a tumor inhibition rate of 27.8%, whereas no inhibitory effect could be seen for Pam3 or SFV/Luc alone. Remarkably, the treatment of tumors with SFV/IFNg alone (inhibitory rate 49.1%) was more efficient than the treatment with SFV/Luc+Pam3 (inhibitory rate 27.8%); thus, the treatment with Pam3 promoted a weak antitumor effect of the control SFV/Luc virus and strengthened the effect of IFNg-expressing SFV/IFNg vector, while Pam3 on its own had no antitumor effect. Altogether, this indicate the efficacy of antitumor treatment mediated by IFNg.

### 3.6. Analysis of Immune Cell Composition of the Tumors Treated with SFV/IFNg

Next, we proceeded to the characterization of the therapeutic potential of IFNg delivered by replication-deficient SFV vector. For this purpose, the tumors raised in mice receiving SFV/IFNg, SFV/Luc, or PBS but no Pam3 (Figure 6d) were subjected to flow cytometry analysis to dissect the composition of tumor-infiltrating immune cells. For this, the tumors were homogenized and stained with lymphoid and myeloid cell markers. We started the analysis with a forward scatter (FSC) and side scatter (SSC) gate to perform a preliminary identification of distinct cell populations [55]. As shown by the light scattering analysis, four common populations distributed according to the cell size and granularity were identified (Figure 7a). Tumors predominantly contain the following distinct cell populations: P1—cancer cells, endothelial cells, fibroblasts, different types of myeloid cells; P2 population—small agranular cells, typically related to T lymphocytes; P3—monocytes; P4—granulocytes. We found that the P2 population was significantly (*p* < 0.01) increased in the SFV-treated tumors compared to the PBS-treated tumors (Figure 7b,c).

Analysis of lymphocytes (CD3^+^) revealed a significant increase in the percentage of CD4^+^ (*p* = 0.0079) among total tumor cells in the group treated with the SFV/IFNg compared to the PBS group, while percent of T-lymphocytes (CD3^+^) did not differ (Figure 8a,b). Remarkably, CD8-positive cells increased in both groups treated with SFV vectors compared to the PBS group (*p* = 0.0476 for both) (Figure 8). Furthermore, the decrease in the T-reg population (CD25^+^/FoxP3^+^/CD4^+^) was characteristic of the group treated with IFNg compared to the SFV/Luc group (*p* < 0.0040) and the PBS group (*p* < 0.0278; Figure 8). These results clearly demonstrated that tumor treatment with the SFV vector led to increased Th and CTL cell populations within the tumor; furthermore, the intratumoral expression of IFNg downregulates the representation of tumor-promoting T-regs in the CD4^+^ population.

The analysis of myeloid cell populations in vivo is not a trivial task because these cells are highly heterogeneous and express overlapping markers at various stages of maturation. The basic phenotypic classification of myeloid cells is based on the CD11b surface marker, which is highly expressed on myeloid cells, including tumor-associated macrophages (TAMs). We observed a significant decrease in CD11b^+^ cells in the SFV/IFNg-treated tumors compared to those in the PBS group (*p* = 0.004) (Figure 9a). The SFV/Luc group revealed a high variability of CD11b^+^ cells. Still, there was a positive correlation between CD11b% and tumor weight (Pearson’s correlation coefficient r = 0.9249; *p* = 0.0244), indicating that small tumors are characterized by low number of CD11b-expressing myeloid cells (Appendix A). Similar results were obtained for the M1 co-injected tumors (Figure 5a). Interestingly, the CD11b^high^ population within CD11b^+^ cells was decreased both in the groups treated with SFV/IFNg and with SFV/Luc compared to the PBS group (although the latter with lower significance; *p* = 0.0278 and *p* = 0.0476, respectively) (Figure 9a). We also observed a decrease in protumorigenic CD11b^+^/CD206^+^ M2 macrophages in the SFV-treated tumors compared to the PBS-treated tumors (*p* = 0.0040 for both SFV/IFNg and SFV/Luc; Figure 9a). The CD11b^+^/CD206^+^ cells were highly positive for Arginase 1, confirming their M2 phenotype (Appendix A). The percentages of CD206^+^ cells in the CD11b cell population did not change (Appendix A).

CD11b^+^/MHCII^high^ cells are considered to be antigen-presenting cells. We did not observe any significant differences in the MHCII^high^ cell population among either the total or CD11b^+^ cell populations (Figure 9b). At the same time, we observed a significant decrease in the CD11b-positive–MHCII-negative population in the tumors treated with SFV/IFNg compared to that of the PBS group (*p* = 0.0476; Figure 9b), which indirectly indicated a decrease in the population of undifferentiated myeloid cells.

In terms of the analysis of the cell size and granularity of myeloid cells, we did not observe differences in the SSC-A/FSC-A populations (P1, P3, P4, Figure 7) between groups of mice, except that in the virus-treated tumors, a significant predominance of MHCII^+^ cells in the P3 population was revealed (not shown). Nevertheless, in total tumor cells, the MHCII marker did not change between groups and populations. Interestingly, MHCII was predominantly found in the P4 population (up to 90% of CD11b^+^ cells were MHCII positive in P4), CD206^+^ cells andArginase1^high^ cells were concentrated in the P1 population, and Arginase1^high^ cells were also found in high amounts in P4, thereby forming two distinct populations (P1 and P4), which correlated with the distribution of CD11b^+^ cells (predominantly found in P1 and P4) (not shown). The CD38 and iNOs markers were widely distributed among the SSC-A/FSC-A populations. As expected, the P2 population did not contain CD11b^+^ or MHCII^+^ cells.

CD38 is a marker that was recently proposed for immunophenotyping of M1 macrophages [56,57]. This finding is highly specific for in vitro studies of BMDM polarization to M1, as we also confirmed in this study (Figure 2); however, in tumors used for in vivo TME characterization, the literature largely indicates an immunosuppressive role of the CD38 marker [58,59,60,61], which has been shown to be associated with myeloid-derived suppressor cells (MDSCs) and T-regs. In that context, we do not consider tumor CD11b^+^/CD38^+^ cells as M1 macrophages. While the impact of CD38 on macrophage phenotyping in vivo remains to be elucidated, we still included the CD38 marker in our antibody panel to characterize the modulation of its intratumoral expression by SFV vector treatment. We observed a decrease in CD11b^+^/CD38^+^ cells in the tumors treated with the SFV vector (Figure 9c). Moreover, the expression of IFNg led to a more significant decrease in the CD11b^+^/CD38^+^ population (SFV/IFNg compared to the PBS group *p* = 0.0040; SFV/Luc compared to the PBS group *p* = 0.0198). We observed a decrease in the expression of the CD38 marker in both CD11b^+^ and CD11b^−^ populations of SFV/IFNg-treated tumors compared to PBS tumors (Figure 9c). Since CD38 is also known to be expressed on nonmyeloid cells, including T-regs [61], a decrease in the CD11b^−^/CD38^+^ population in SFV treated tumors could be associated with a decrease in the populations of T-regs (Figure 8a).

Finally, we analyzed the expression levels of Arginase 1 and iNOs in the CD11b^+^ population as immunosuppressive (Arginase 1) or inflammatory (iNOs) markers, respectively. In general, we did not find differences in the total expression levels of these enzymes between the groups; however, surprisingly, Arginase 1^+^/iNOs^−^ cells increased in the CD11b^+^ population in the tumors treated with SFV/IFNg (PBS compared to SFV/IFNg *p* = 0.0278; SFV/Luc compared to SFV/IFNg *p* = 0.0476) (Figure 9d). Furthermore, iNOs^+^/Arginase 1^−^ cells increased in the CD11b^+^ population of tumors treated with SFV/IFNg (PBS compared to SFV/IFNg *p* = 0.0476). Although there were no differences in the iNOs^+^/Arginase1^−^ cells between the SFV/IFNg and SFV/Luc groups, interestingly the smallest tumor in the SFV/Luc group demonstrated the highest number of iNOs^+^/Arginase 1^−^ cells (Pearson’s correlation coefficient r = −0.8407; *p* = 0.0372) (Appendix A).

In summary, we concluded that tumor treatment with SFV/IFNg led to an increase in Th and CTL cells and a decrease in T-regs in the CD4^+^ cell population. Furthermore, treatment with SFV/IFNg inhibited CD11b^+^ cell infiltration and decreased the CD206^+^ and CD38^+^ cell populations, explaining the observed inhibition of tumor growth.

## 4. Discussion

In this study, we have evaluated the therapeutic potential of the SFV/IFNg vector in a three-dimensional (3D) in vitro system and in a mouse breast cancer model in vivo. Currently, there is no reliable 3D model that can be used to investigate the interplay between cancer cells, immune cells, and viral vectors in vitro. We designed and for the first time tested the cancer cell spheroid-based model for SFV vector delivery of IFNg, as well as its ability to activate macrophages under free-floating conditions.

Similar to many other cancer cells, 4T1 cells can form spheroids in nonadherent conditions [62]. As shown in several studies, 3D cultivation results in physiologically and (epi)genetically relevant features of solid tumors, including different zones of proliferation, an oxygen gradient, a natural extracellular environment, an increased stemness-related gene expression pattern, and stimulation of epithelial to mesenchymal transition-related gene expression [63,64,65]. Previously, numerous attempts were made to establish a 4T1-based 3D/spheroid tumor model. The first one employed coculture of 4T1 cells with murine embryonic fibroblasts on Matrigel [66]. Tumor-surrounding fibroblasts played a role in distributing and connecting epithelial breast cancer cells to mimic the tumor microenvironment [66]. In this, and in settings using an alginate matrix, coculturing with fibroblasts (especially, NIH/3T3 cells) significantly supported the proliferation, scattering, and invasiveness of 4T1 cells [67,68]. There are also other systems used for culturing cells in spheroids, using other hydrogels, diverse scaffolds, or the hanging drop method [69]. Here, we for the first time established a reproducible and relatively simple 3D spheroid model based solely on 4T1 cultured without a scaffold or hydrogel support, to further use it to study the susceptibility of 3D-cultured tumor cells to immunotherapy with activated macrophages and viral vectors.

The main problems with viral-based therapy are the low efficacy of vector delivery and poor distribution within the tumor. Indeed, while oncolytic viruses are highly efficient in killing tumor cells in vitro in 2D monolayers, their efficiency is significantly lower in 3D environments, both in vitro and in vivo. Due to this, prior to in vivo application, the virotherapy has to be pretested in the 3D tumor cell–culture systems. The spatial dimensions in the spheroid allow mimicking of the dynamics of virus spread in the tumor, and through this treatment optimization [70,71]. Here, for the first time we applied a spheroid-based cancer model to characterize the antitumoral activity of virotherapy with Semliki Forest Virus vectors made to encode IFNg (SFV/IFNg). In general, the spheroid system established here allows real-time monitoring of spheroid infection and virus distribution as a small 3D tumor model. Infection of the 4T1 spheroids with SFV/DS-Red virus clearly showed the limited distribution of the virus within the spheroid (Figure 1), which in general reflects the in vivo conditions.

The addition of immune cells to the 3D system is a prerequisite for in vitro TME modeling. We added BMDMs to the infected spheroids to evaluate the effect of virus-derived IFNg (vdIFNg) on spheroid growth. As shown in Figure 2, vdIFNg in the presence of TLR2/1 ligand Pam3 efficiently activated macrophages to the M1 phenotype in 3D plates, demonstrating a marker profile similar to that of monolayer conditions. We expected that nitric oxide (NO) produced by activated macrophages would inhibit the growth of 4T1/eGFP spheroids; furthermore, IFNg was shown to have a direct antiproliferative effect on cancer cells [72,73], providing strong evidence of spheroid inhibition in the presence of M1 and vdIFNg. As expected, the highest inhibitory effect was observed in the spheroids infected with SFV/IFNg in the presence of M0 macrophages and Pam3 at day 10 (Figure 3b). Surprisingly, SFV/Luc infection also inhibited spheroid growth in the presence of M0 macrophages. This indicated possible sensitization or polarization of macrophages by SFV/Luc + Pam3 (without IFNg). Importantly, Pam 3 on its own did not affect macrophage polarization or 4T1 cell growth under 2D and 3D conditions. Nevertheless, sensing of macrophages by SFV/Luc + Pam3 without vdIFNg or in the presence of recombinant “pure” IFNg (nonviral) may have potential for the analysis of the possible synergy of the antiviral immune response and M1 polarization, which is the subject of further studies.

Cell infection with alphaviruses induces a type I IFN response and results in the expression of other cytokines and chemokines by infected cells [74,75]. Macrophages express receptors for all three types of IFNs, which stimulate the expression of hundreds of genes known as IFN-stimulated genes (ISGs). Although BMDMs are not activated in vitro without IFNg, the presence of IFNα/β can potentially stimulate the inflammatory response in macrophages [76], which would explain the inhibitory effect of SFV/Luc+M0. This is an interesting scenario, as it shows that tumor virotherapy with SFV could be achieved through the induction of antiviral innate immune response leading to M1 polarization, opposing the immunosuppressive protection of virus replication at the cost of hindering the antitumor immune response [77].

Interestingly, we also observed inhibition of spheroid growth by M0 alone (sph+M0). Importantly, this inhibition was observed at very late stages of spheroid cultivation (day 10). The total number of cells (dividing 4T1 cells and slow-dividing macrophages) or cell density at this stage is relatively high, which limits spheroid growth. Growth inhibition could be caused by limitations in the resources of the cell medium. The supply or access to oxygen, in particular, inside the spheroids can be critical. In this respect, 3D spheroid culture may mimic the intratumoral hypoxia. Cycling or intermittent hypoxia occurs in solid tumors and affects different cell types in the tumor microenvironment, and in particular the tumor-associated macrophages (TAMs). Interestingly, it was found to modulate the phenotype of TAMs, specifically to polarize unpolarized (M0) murine BMDM to the M1 phenotype characterized by an increase in the secretion of TNFα and IL-8/MIP-2. The pro-inflammatory phenotype of M1 macrophages induced by hypoxia was evidenced by increased pro-inflammatory cytokine secretion and pro-inflammatory gene expression [78]. Our data indicated that this effect could be mimicked in the 3D culture at the late stages of spheroid growth, and that the emerging population of M1-like cells can partially inhibit spheroid growth.

The unexpected inhibition of spheroid growth by M0 macrophages prompted us to estimate the impacts of M0 and polarized M1 (vdIFNg) macrophages on tumor growth in vivo upon coinjection of macrophages with 4T1 (Luc2) tumor cells. Similar coinjection experiments were performed to evaluate the role of M2 macrophages prepolarized in vitro by IL-4 [79,80]. Although we observed a significant decrease in tumor weight in the 4T1(Luc2)+M1 group, tumor growth inhibition was detected only on the early days of monitoring the bioluminescence signal (Figure 4). Furthermore, we did not observe inhibition of lung metastasis in the M1 group, indicating an insufficient therapeutic potential of M1 alone. It is widely accepted that M1 macrophages have antitumorigenic functions, whereas M2 and M0 macrophages exhibit a tumor-promoting phenotype [81]. As demonstrated in Figure 2, macrophages polarized to M1 represent a heterogeneous population because not all cells express M1 markers and most cells probably remain M0. We can assume that these cells are partially M1-potentiated. Furthermore, high M1 plasticity was confirmed by recent studies [82,83]; therefore, elimination of the M1 stimulus (vdIFNg) may lead to reversible reprogramming of M1 to a tumor-promoting phenotype in vivo. Nevertheless, M1 coinjection inhibited tumor weight and affected myeloid and lymphoid cell subsets by inhibiting CD11b^+^ cell infiltration and decreasing the number of T-regs in the TME (Figure 5), demonstrating the therapeutic potential of the M1 polarization strategy by vdIFNg.

Intratumoral administration of viral vectors expressing IFNg represents a promising strategy for immunomodulation of the TME, especially for locally advanced tumors (breast cancer, prostate cancer), which allows i.t. administration to avoid the systemic toxicity of IFNg [84]. In this study, the subcutaneous model revealed a high diversity of tumor growth within each group and the absence of significant inhibition of treated tumors, in contrast to the orthotopic model (Figure 6). Subcutaneous and orthotopic models possess different biologic parameters related to tumor perfusion efficacy, hypoxic burden, microvasculature density, and immune cell infiltration, which is crucial to the immunotherapy outcome. Orthotopic tumors usually exhibit increased malignant behavior and less variability [85]. We revealed significant tumor inhibition specifically in an orthotopic model, confirming the potential of the proposed SFV/IFNg vector for primary breast cancer immunotherapy.

Both treatments, the SFV/IFNg vector alone or in combination with TLR2/1 Pam3 ligand, revealed inhibition of tumor growth in the orthotopic model. Moreover, SFV/Luc in combination with Pam3 inhibited tumor growth compared to that of the PBS group. TLR agonists are known immunological adjuvants for cancer therapy and anticancer vaccines [86,87]. Pam3 was shown to reduce the suppressive function of T-regs and enhance the cytotoxicity of tumor-specific CTLs [88,89,90]. It can also inhibit tumor growth on its own [53]. Nevertheless, in our study i.t. Pam3 injection had no effect on tumor growth (Figure 6c). We can speculate on the synergistic effect of TLR2 agonists and SFV vectors; however, a more detailed dose-dependent Pam3 study is required to demonstrate potential synergy.

Comparison of intratumoral immune cell infiltrates revealed a significant increase in T cell populations in the tumors treated with SFV/IFNg. In previous studies, IFNg treatment increased the proliferation of CD4+ Th cells [91] and promoted the IFNg-dependent infiltration of T-cells into tumors [92]. We observed significant increases in the CD8 cell populations in the SFV/IFNg- and SFV/Luc-treated tumors and in CD4 cells in the SFV/IFNg-treated tumors (Figure 8). CD8 cell recruitment and activation is a characteristic feature of virus-based therapy approaches [93,94]. This dual effect of virus replication and IFNg as a CD4 effector molecule synergizes with the therapeutic outcome in treated mice. Furthermore, the decrease in the T-reg population within CD4+ cells is a complementary component of tumor growth inhibition. Low numbers of T-regs and high CD4+ and CD8+ cell/T-reg ratios are considered good prognostic factors [95,96]. IFNg can induce the fragility of tumor-derived T-regs, the loss of T-reg suppressive activity [97,98], and the inhibition of T-reg expansion [99]. These data support our results showing T-reg inhibition in the SFV/IFNg-treated tumors, in contrast to SFV/Luc virus treatment, which induced only an increase in the number of CD8 cells. Interestingly, M1 coinjection with cancer cells also resulted in decreased T-reg cell populations in CD4+ cells (Figure 5), which may have been related to IFNg-based macrophage activation and downstream M1–T cell crosstalk through IFNg/IL-12 signaling [100].

In addition to lymphocytes, myeloid cells play an important role in TME programming. Myeloid populations such as tumor-associated macrophages (TAMs), neutrophils, and myeloid-derived suppressor cells (MDSCs) are the most abundant immune cells within tumors. We used CD11b as a typical myeloid lineage marker to characterize the infiltration of these cells in treated tumors. High CD11b^+^ cell infiltration usually correlates with tumor progression, invasion, and metastasis, and patients with high CD11b^+^ cell infiltration have a poorer surgical outcome [101]. In this study, we demonstrated a significant inhibition of the CD11b^+^ cell population in the tumors treated with SFV/IFNg (Figure 9a). Furthermore, the number of CD11b^high^ cells was also lower in the virus-treated tumors. A similar decrease in the CD11b^+^ population was observed in the tumors coinjected with M1 (Figure 5a). The mechanism of the IFNg-attributed decrease in CD11b^+^ cell infiltration is unknown. Although a linkage between the CD11b^+^ decrease, T-reg decrease, and increase in Th and CTL cells is clearly visible, the role of IFNg in immune modulation through targeting of myeloid cells is unclear. It was shown previously that tumor-infiltrating T cells gradually lose their capacity to produce IFNg through post-transcriptional inhibitory events, meaning they fail to clear malignant cells [102]. The exogeneous vector-based production of IFNg may directly stimulate myeloid cell differentiation to a proinflammatory phenotype to restore the IFNg/IL12 axis between M1 macrophages and T cells.

IFNg upregulates the expression of antigen presentation molecules, both MHC I and MHC II, stimulating the CTL response against cancer cells and an inflammatory Th1 adaptive response [103,104]. In vitro in a 3D system, we observed a significant increase in MHCII in the BMDMs treated with vdIFNg/Pam3. A relatively high level of MHCII is classically observed on M1 macrophages. Surprisingly, we did not find any significant differences in MHCII^high^ expression on CD11b^+^ cells or CD11b^−^ cells in the analyzed tumors. Nevertheless, the numbers of CD11b^+^/MHC II^−^ cells decreased significantly in the SFV/IFNg-treated tumors (Figure 9b), indicating possible differences in specific myeloid cell populations, which were not analyzed in this study. More detailed analysis of MHC II levels within specific myeloid cell subsets is necessary to fully characterize the effects of SFV/IFNg treatment on MHC II levels and to unravel the mechanism vdIFNg stimulation of lymphoid and myeloid cells in the TME.

Finally, CD206 and Arginase 1 markers, which are associated with the M2-like protumorigenic phenotype [50], as well as the CD38 marker, which is expressed on immunosuppressive myeloid cell types [58,59,60], were analyzed. In this study, the decrease in CD206^+^ cells in the SFV/IFNg-treated tumors generally was associated with the decrease in CD11b^+^ cells infiltration (Figure 9a). Interestingly, the SFV/Luc vector also inhibited the CD206^+^ population. The virus-based inhibition of M2 can be related to the CD8-mediated response to infected cells. Remarkably, most CD206^+^ cells were highly positive for Arginase 1, confirming the M2 phenotype of these cells. Nevertheless, the analysis of the CD11b^+^ population revealed increases in both Arginase 1 and inducible NO synthase (iNOs) in the tumors treated with SFV/IFNg (Figure 9d), which could be attributed to the increased expression of these markers by other cell populations (than monocytes/macrophages), such as neutrophils, indicating the complexity of the effects of SFV/IFNg vector treatment on the TME, stretching beyond the macrophages [22,105].

CD38 is expressed across different immune cell subsets, including T cells, myeloid cells, NK cells, and B cells. Recently, CD38-related immunosuppression was attributed to T-regs and MDSC populations [60,61,106]. In this study, strong inhibition of CD11b+/CD38+ (provisional MDSCs) was revealed in the tumors treated with SFV/IFNg. Interestingly, CD11b-/CD38+ (nonmyeloid) cells were also decreased in the SFV/IFNg group, indicating total CD38 marker inhibition, which can be related to the observed inhibition of T-regs, as discussed above. Although the analysis of CD38 cannot directly confirm the impact of SFV/IFNg on MDSCs, the total decrease in CD38 can be considered as an important therapeutic indicator of the SFV/IFNg treatment.

The role of IFNg in the activation of myeloid-derived cells in vivo and in vitro is controversial and not completely clear. Nevertheless, the beneficial antitumor effect of SFV/IFNg shown in this study may contribute to establishing promising immunotherapies for cancer in the future. The intratumoral expression of IFNg has the potential to reprogram the TME by decreasing the populations of intratumoral T-regs and myeloid cells, as well as by activating the antitumor T cell subsets, which is enhanced by the antiproliferative qualities of IFNg in synergy with the induction of apoptosis in SFV-infected cancer cells. The application of this vector for therapeutic tuning of the TME represents a very promising strategy for the development of combined immunotherapy–chemotherapy treatments targeting different tumor escape pathways.

## 5. Conclusions

In this study, we have shown that the SFV/IFNg vector inhibits tumor growth in an orthotopic 4T1 mouse breast cancer model. In tumors treated with SFV/IFNg, the inhibition was related to significant increase in the populations of intratumoral Th cells and CTLs, and reduction in the share of T-regs within the Th populations. Furthermore, tumor growth inhibition was associated with decreased intratumoral infiltration of myeloid cells expressing CD11b, CD206, or CD38. SFV-based expression of IFNg benefits the antitumor immune response, representing a promising adjuvant to current immunotherapy and chemotherapy strategies.

Additionally, we developed a method for coculturing cancer cell spheroids and macrophages under free-floating 3D conditions to investigate the SFV-based delivery of IFNg and to decipher the direct macrophage-inhibitory effects on cancer spheroid growth. This method can facilitate various cancer research and treatment approaches and can be useful for modeling the virus-based delivery of immune-modulating genes in the presence of free-floating immune cells under 3D conditions.

## Figures and Tables

**Figure 1 vaccines-09-01247-f001:**
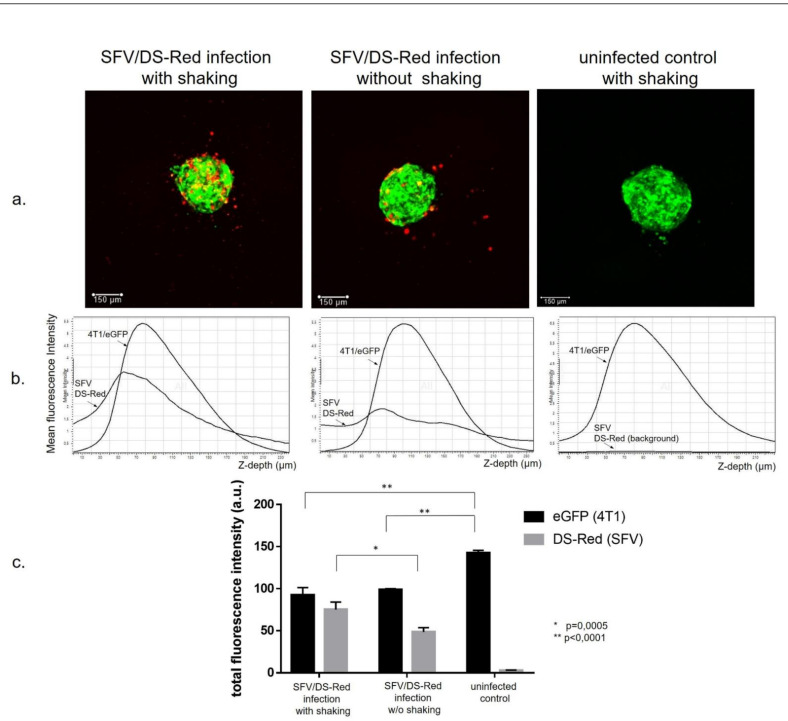
Infection of the 4T1/eGFP cancer cell spheroids with recombinant SFV/DS-Red virus. The 4T1/eGFP cells (3 × 10^3^ cells per well) were plated into 96-well ultralow attachment plates. The next day, the spheroids were infected with SFV/DS-Red virus (10^5^ i.u./well) either with or without shaking during incubation with the virus (1 h 10 min). Then, the spheroids were incubated for 2 days to allow DS-Red transgene expression. Confocal microscopy was performed using a Leica TCS SP8 microscope, and the images were processed by LasX software as described in the methods. (**a**) Maximum intensity projection confocal images of the 4T1/eGFP spheroids infected with SFV/DS-Red virus (representative images). (**b**) Graphical analysis of the mean fluorescence intensity changes from the spheroid upper rim to the spheroid bottom. The eGFP and DS-Red signal curves are indicated by arrows; 48 planes, z step 5 µm. (**c**) Total fluorescence intensity of the spheroids incubated with or without shaking during infection and the uninfected spheroid controls. Data are presented as the mean of total fluorescence of four spheroids in each group. Error bars represent the standard deviation, *n* = 4. Statistical analysis was performed by Tukey’s multiple comparison test.

**Figure 2 vaccines-09-01247-f002:**
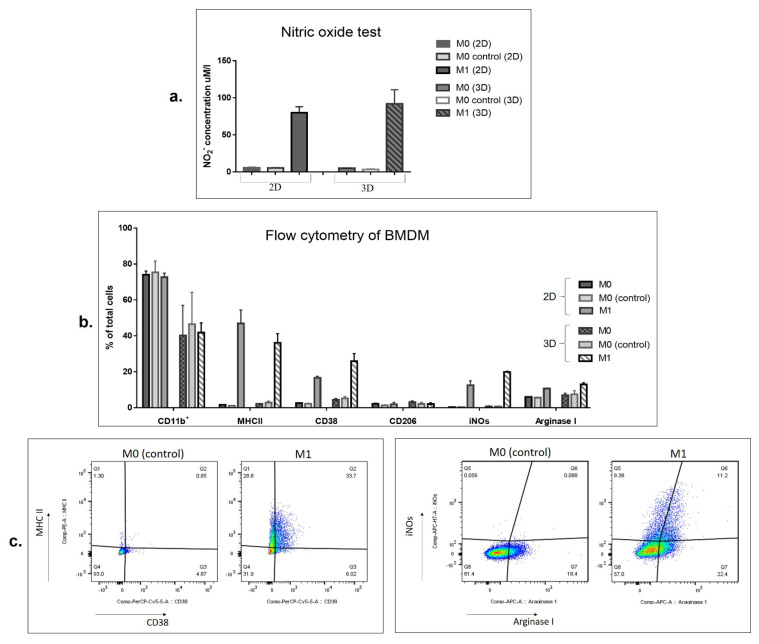
SFV virus-derived IFNg (vdIFNg) activates BMDMs to the M1 phenotype in monolayers (2D) and under free-floating conditions (3D). BMDMs (M0) were seeded in 12-well plates (2D) and in 96-well ultralow attachment plates and incubated for 2 days in the presence of 50 ng/mL vdIFNg and 100 ng/mL Pam3 to polarize macrophages to an M1-like phenotype (M1). M0 control represents BMDMs incubated with vdLuc supernatant (SFV/Luc conditioned medium) obtained in a similar manner as vdIFNg by infection of BHK-21 cells with the respective virus (SFV/Luc, SFV/IFNg). M0 represents untreated BMDMs. (**a**) Production of nitric oxide (NO) by macrophages activated to M1 under 2D and 3D conditions. The level of nitric oxide was determined in cell culture supernatants after 2 days of BMDM activation with vdIFNg under two different conditions (2D, 3D). (**b**) Flow cytometry analysis of macrophage surfaces and intracellular markers after 2 days of activation in 2D and 3D conditions. The diagram shows % of total single cells, the bars represent the mean values ± SD. (**c**) Representative images of flow cytometry illustrating the increase in the MHCII, CD38, and iNOs markers in M1 macrophages under 3D cultivation.

**Figure 3 vaccines-09-01247-f003:**
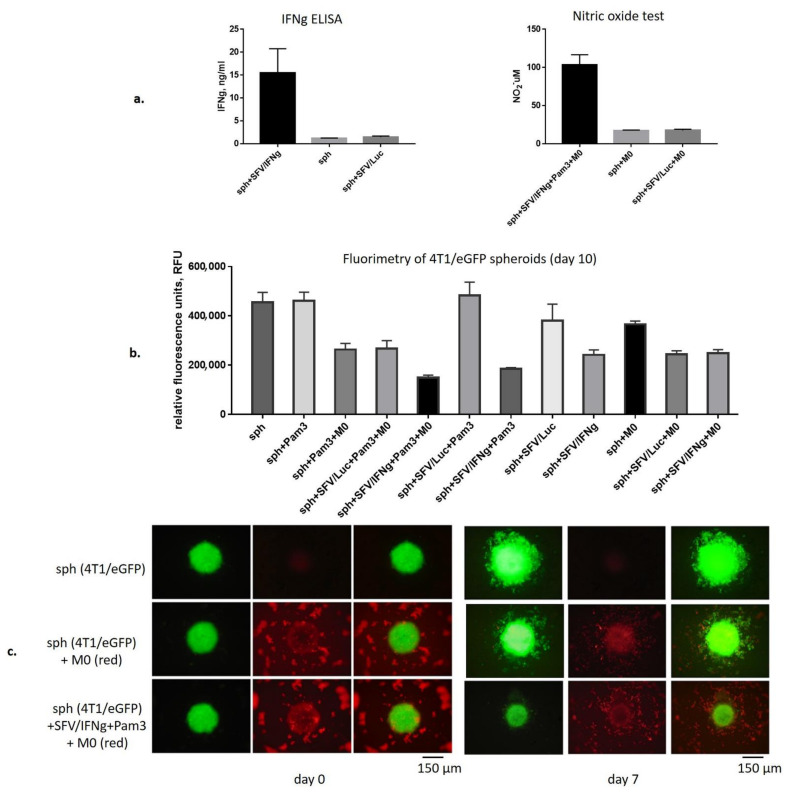
Coculturing of SFV/IFNg-infected 4T1/eGFP spheroids with macrophages. Single 4T1/eGFP spheroids were generated from 3000 cells in 96-well ultralow attachment plates. The next day, the spheroids were infected with either SFV/IFNg or SFV/Luc (5 × 10^4^ i.u./well) or incubated with PBS as the uninfected control. The next day after infection, BMDMs (3 × 10^4^ cells/well) were added to the spheroids (+M0, day 0). In total, twelve combination groups (six single spheroids in each group, *n* = 6) were prepared: sph—uninfected spheroids (PBS); sph+Pam3; sph+Pam3+M0; sph+SFV/Luc+Pam3+M0; sph+SFV/IFNg+Pam3+M0; sph+SFV/Luc+Pam3; sph+SFV/IFNg+Pam3; sph+SFV/Luc; sph+SFV/IFNg; sph+M0; sph+SFV/Luc+M0; sph+SFV/IFNg+M0. Pam3 was added to a final concentration of 100ng/mL to respective groups. (**a**) The production of vdIFNg by spheroids was measured in cell culture supernatants by ELISAs 18 h after infection, before the macrophages were added. The production of NO by macrophages was measured in cell culture supernatants after two days of incubation with infected spheroids. (**b**) The total eGFP fluorescence measured by fluorimetry at day 10 of the incubation. (**c**) Representative fluorescence microscopy images of spheroids incubated with prestained macrophages (red) at day 0 and day 7. Bars represent the mean values ± SD, *n* = 6.

**Figure 4 vaccines-09-01247-f004:**
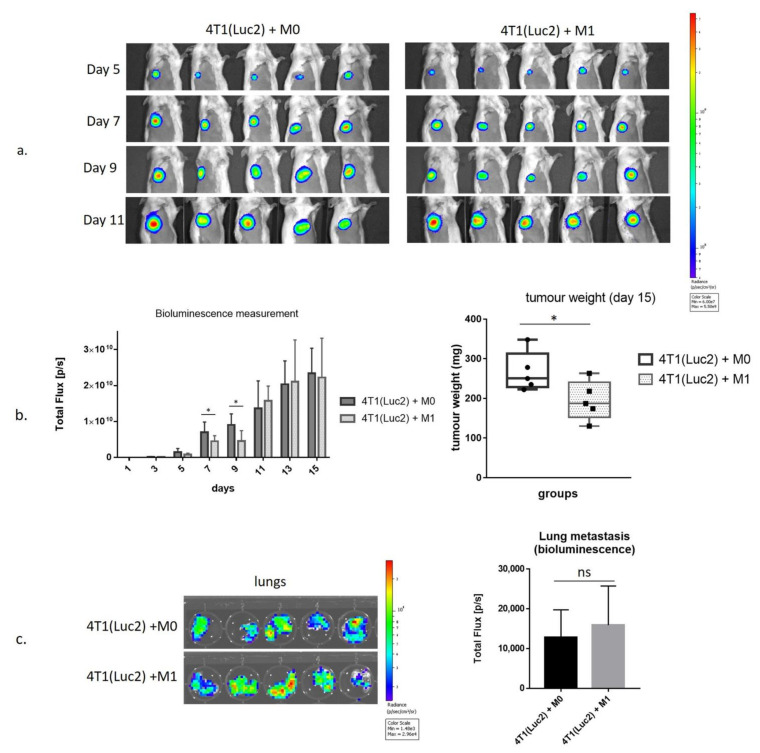
The effect of M1 macrophages polarized by SFV-derived IFNg on 4T1(Luc2) tumor growth. The 4T1(Luc2) cells (1 × 10^4^) were orthotopically coinjected with 2 × 10^4^ M0 (4T1(Luc2)+M0) or M1 (4T1(Luc2)+M1) macrophages in BALB/c mice (*n* = 5 per group) at day 0. (**a**) In vivo bioluminescent imaging of 4T1(Luc2) tumors expressing luciferase (days 5–11). (**b**) Quantitative analysis of tumor bioluminescence and the tumor weights. (**c**) Bioluminescence imaging of lungs isolated from the tumor-bearing mice. Bars represent the means ± SD (*n* = 5); *—significant difference (*p* < 0.05); ns—nonsignificant.

**Figure 5 vaccines-09-01247-f005:**
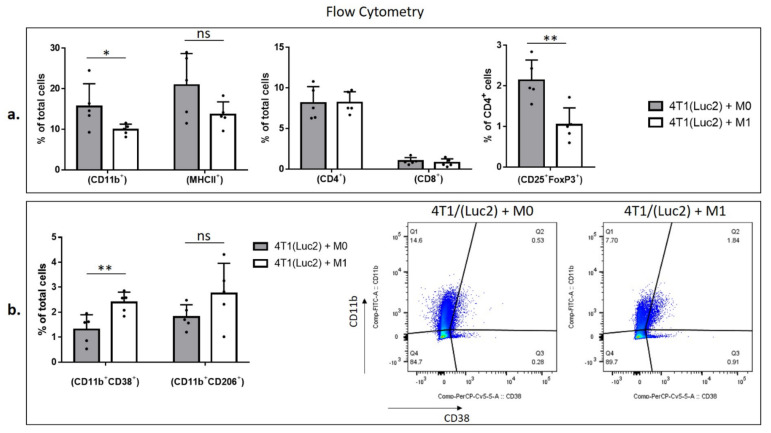
Flow cytometry analysis of immune cells isolated from tumors generated by implantation of 4T1(Luc2) cells premixed with M0 or M1 macrophages. Tumors were homogenized and a single cell suspension was used for immunostaining (see Materials and methods for the details). Flow cytometry was performed to quantify the immune cell populations (%): (**a**) CD11b^+^; MHCII^+^; CD4^+^; CD8^+^ in total single cells and CD25^+^/FoxP3^+^ in the CD4^+^ population; and (**b**) double positive CD11b^+^/CD38^+^, CD11b^+^/CD206^+^ cells in total single cells. Representative flow cytometry gating data of CD11b^+^ and CD38^+^ cells are shown. Bars represent the means ± SD (*n* = 5); * *p* < 0.05; ** *p* < 0.01; ns—nonsignificant.

**Figure 6 vaccines-09-01247-f006:**
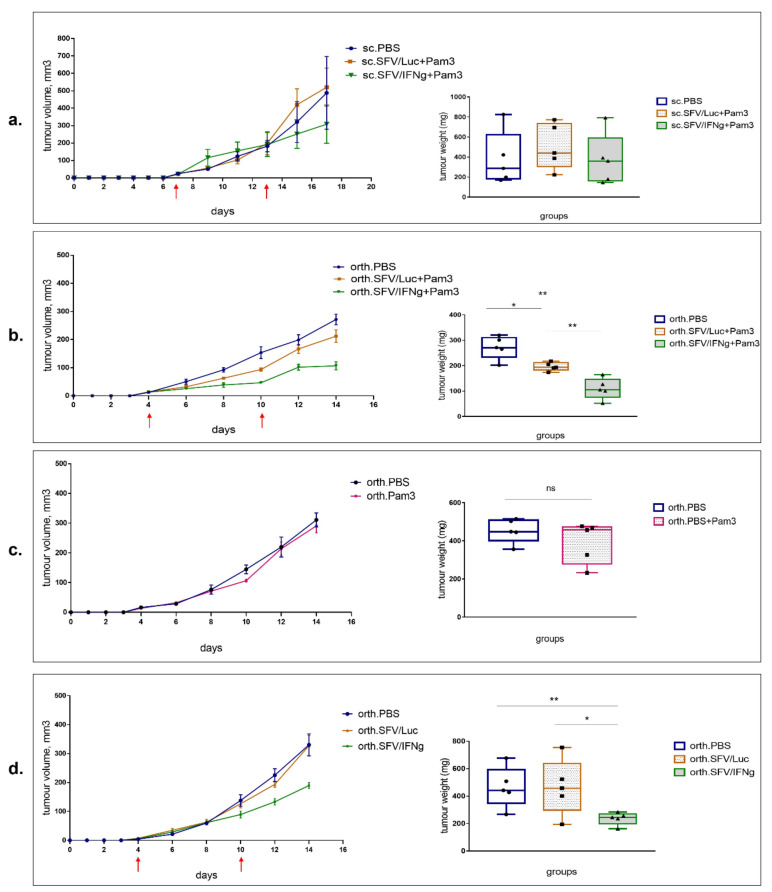
Inhibition of 4T1 tumor growth by i.t. injection of SFV/IFNg virus. The 4T1 mouse breast tumors were established by subcutaneous (sc.) or orthotopic injections of 2.5 × 10^5^ and 1.25 × 10^5^ 4T1 cells, respectively. Mice received two i.t. injections of SFV vectors (4 × 10^7^ i.u./tumor) or PBS control, as indicated by red arrow time points. The tumor growth curves are shown on the left and the tumor weights of the groups are shown on the right. (**a**) Treatment of subcutaneous tumors with SFV/IFNg, SFV/Luc, and PBS (control). The next day after virus administration, mice in these groups received i.t. injections of Pam3 solution (10 µg/tumor at day 8 after first virus administration and 15 µg/tumor at day 14 after second virus administration). (**b**) Treatment of orthotopic tumors with SFV/IFNg, SFV/Luc, and PBS (control). The day after virus administration, mice in these groups similarly received i.t. injections of Pam3 solution (10 µg/tumor at day 5 after first virus administration and 15 µg/tumor at day 11 after second virus administration). (**c**) Treatment of orthotopic tumors with only Pam3 (or PBS) solution: first Pam3 (10 µg/tumor) i.t. injection at day 5; second Pam3 i.t. injection (15 µg/tumor) at day 11. (**d**) Treatment of orthotopic tumors only with SFV/IFNg, SFV/Luc, and PBS (control), without the subsequent Pam3 injection. Bars represent the means ± SD (*n* = 5); * *p* < 0.05; ** *p* < 0.01; ns—nonsignificant.

**Figure 7 vaccines-09-01247-f007:**
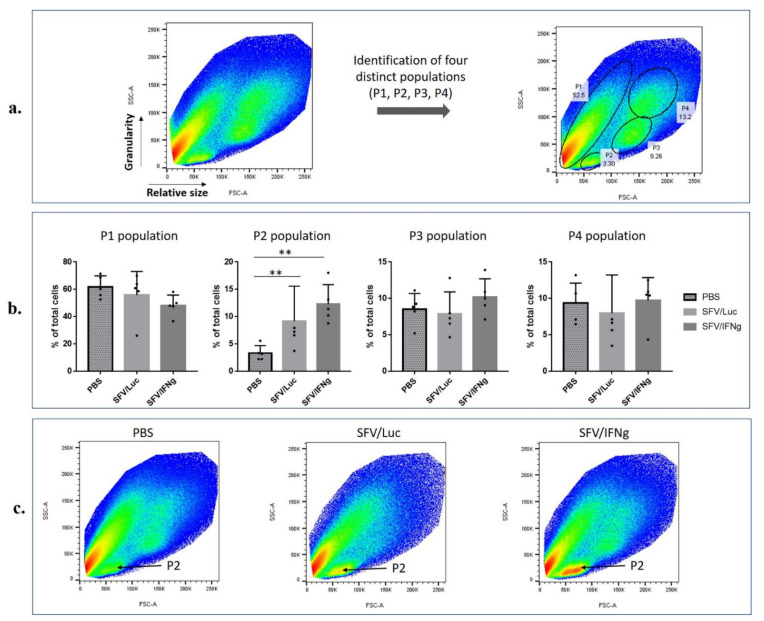
Flow cytometry of the 4T1 tumors treated with SFV/IFNg, SFV/Luc or PBS (mice were treated as presented in Figure 6d). Tumors were resected, homogenized to obtain a single cell suspension, and the total isolated cells were subjected for immunostaning followed by analysis of forward and side (SSC-A/FSC-A) scattering of cell populations. (**a**) Schematic representation of the identification of four distinct single-cell populations (representative pictures). (**b**) Percentage of the respective populations within total single cells (five mice per group). (**c**) Representative SSC-A/FSC-A data from the PBS, SFV/Luc, and SFV/IFNg groups. The pronounced P2 populations are indicated by arrows. Bars represent the means ± SD (*n* = 5); ** *p* < 0.01.

**Figure 8 vaccines-09-01247-f008:**
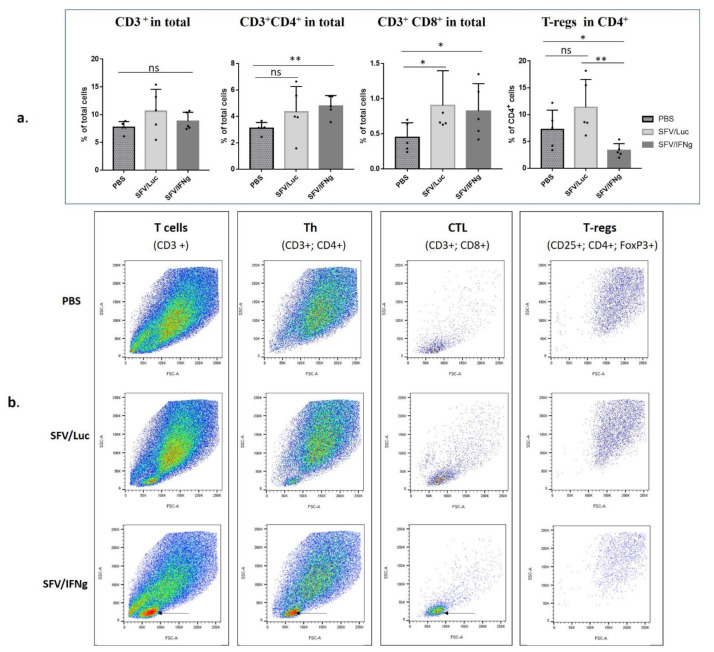
Flow cytometry analysis of T cells in the tumors treated with SFV/IFNg, SFV/Luc, or PBS in orthotopic 4T1 mouse breast cancer model (mice were treated as presented in Figure 6d). Resected tumors were homogenized to obtain a single cell suspension, which was used for immunostaining with antibodies against surface markers (CD3, CD4, CD8, CD25) and intracellular markers (FoxP3) in one mixture. (**a**) Percentages of cell populations. Th cells were identified as CD3^+^/CD4^+^; CTL cells as CD3^+^/CD8^+^, and T-regs as CD4^+^/CD25^+^/FoxP3^+^ populations. (**b**) Representative pictures of forward and side (SSC-A/FSC-A) scattering of the cell populations. Arrows indicate the increased population of lymphocytes in the SFV/IFNg group. Bars represent the means ± SD (*n* = 5); * *p* < 0.05; ** *p* < 0.01; ns—nonsignificant.

**Figure 9 vaccines-09-01247-f009:**
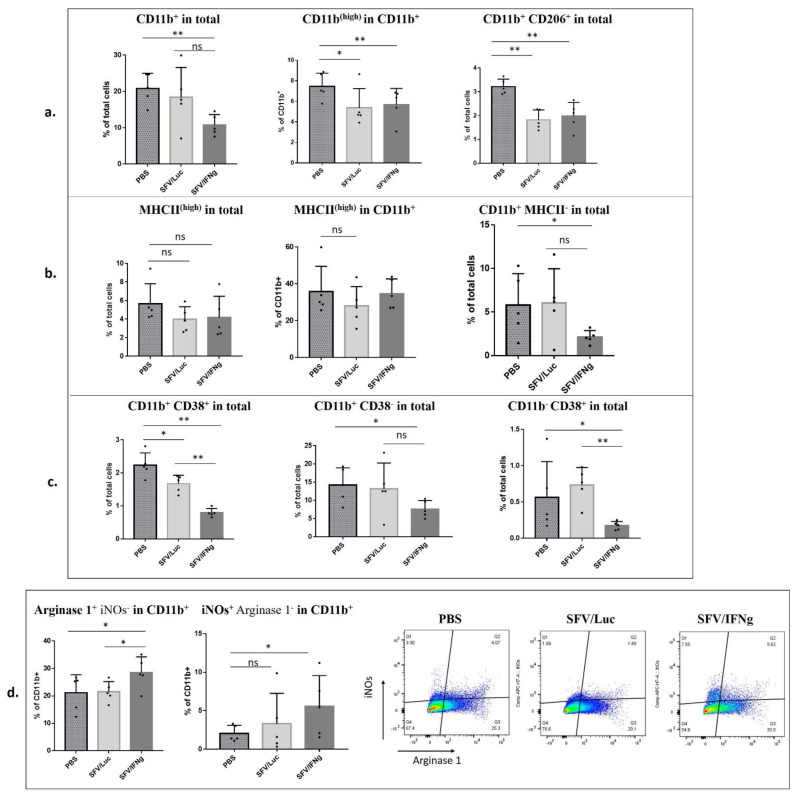
Flow cytometry analysis of myeloid cells in the tumors treated with SFV/IFNg, SFV/Luc, or PBS in an orthotopic 4T1 mouse breast cancer model. Mice were treated as shown in Figure 6d. Resected tumors were homogenized to obtain a single cell suspension, which was used for immunostaining with antibodies against surface markers (CD11b, CD206, MHCII, CD38) and intracellular markers (Arginase 1, iNOs) in one mixture. (**a**) Percentages of surface markers CD11b, CD11b (high), and double-positive CD206/CD11b. (**b**) Percentages of surface markers MHCII (high) and CD11b^+^MHCII^−^. (**c**) Percentages of surface markers CD11b^+^/CD38^+^ and CD11b^−^/CD38^+^ (**d**) Percentages of intracellular markers (Arginase 1; iNOs, inducible NO synthase); representative images of Arginase 1 and iNOs gating in the CD11b^+^ population are shown. Bars represent the means ± SD (*n* = 5); * *p* < 0.05; ** *p* < 0.01; ns—nonsignificant.

## Data Availability

The data presented in this study are available on request from the corresponding author.

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
