# Peer review of "Alphavirus-Driven Interferon Gamma (IFNg) Expression Inhibits Tumor Growth in Orthotopic 4T1 Breast Cancer Model"

_vaccines, 2021, doi:10.3390/vaccines9111247_

Round 1
Reviewer 1 Report
The work of Trafimova and colleagues describes the efficacy of cancer immunotherapy with IFN-g carried by a viral vector in an in vitro model of mouse cancer cell line spheroids and cocultures with in vitro differentiated macrophages, and in orthotopic and subcutaneous 4T1 in vivo mouse models of cancer. The study is original with a certain interest and the experimental models well chosen. In addition, the context is well laid out in the introduction. However, there are some important experimental and analytical points to consider as they can prevent the correct interpretation of the results and thus lead to wrong conclusions.
Importants comments and observations
- There are gaps in the flow cytometry analysis part. Indeed, it is not indicated whether a viability marker was used. This would be particularly essential for the analysis of immune populations in tissues, knowing that the latter contain a large part of necrotic cells which could compromise the stainings, which is more certain during an acquisition which can extend up to 2 days. Furthermore, the fluorochromes of the antibodies used throughout the study are not specified. Finally, the choice to perform the gating according to the unmarked samples is not optimal, in particular when multi-colored panels are used and (fluorescence-minus-one) are then essential. This is therefore felt in Figure 9D, in which the positive/negative limit for the arginase staining is not pushed far enough to the left. Indeed, a normal staining of arginase is supposed to lead to 2 distinct populations which is not evident in the current study. Finally, the raw data presented in Figures 2C and 5B are difficult to exploit because of the too low voltage applied for the FITC and PC5.5 channel. Thus a thorough re-analysis of the cytometry data is necessary.
Minor points
- Line 18: “and in vivo” should be italics
- Line 115: please provide the origin and the sequence species of M-CSF
- Line 127: please confirm if any selection antibiotic has been used for the maintenance of the 4T1-Luc2 cell line
- Line 386: please specifiy the subversion of FlowJo used, e.g. v10.7.2
- Line 422: you claim that the growth of the spheroid is different in uninfected and infected groups, but you should also considere a different viability, i.e. more cell death due to the infection
- Figure 4B (left): you should considere using repeated-measures ANOVA for the statistical analysis of the bioluminescence images
- Figure 6: would you please also show the tumor growth curves for all individual animals?
Reviewer 2 Report
The manuscript by Trofimova et al. is an interesting study showing the antitumor potential of a self-replicating RNA vector based on SFV expressing IFN-gamma. The research focuses on the role played by pro-inflammatory macrophages (M1) in the viral-based treatment. The authors address this by using both 3D spheroids and in vivo mammary tumor models, and treating them with SFV-IFN-gamma alone or combined with a TLR2-1 agonist, which has shown to be able to polarize macrophages to M1 phenotype in the presence of IFN-gamma. This combination showed the best antitumor effects in an orthotopic 4T1 model, which was extensively characterized at the immunological cellular level. The work is quite complete and the 3D in vitro model developed by the authors represents an interesting tool to evaluate the effect of antitumor treatments in animal-free systems, something highly demanded nowadays.
Only some minor points should be addressed
- Fig 1 & 3. These are very nice experiments, showing that 3D spheroids of tumor cells can be infected in vitro and used to analyse antitumor effects. My only critique here is that the spheroids are infected from outside, this is, they are incubated with the virus added to the medium. This is not the same situation in vivo, where the virus is injected intratumorally. Would it be possible to microinject the 3D spheroids in vitro with SFV? Have the authors attempted that?
- Supplementary Figures: legends for supplementary figures are too brief, please elaborate them a bit more.
- Supplementary Fig.S1c. The authors measured luciferase expression from spheroid lysates, but in the main text it is indicated that spheroids used in that particular experiment expressed GFP. So, it is not clear how this experiment was performed. It is possible that the authors used spheroids expressing luciferase for this experiment. In that case, how did they distinguish between tumor cell-luciferase and luciferase expressed by SFV-Luc?
- Fig 6. In the legend for this figure it is stated that Pam3 solution was given as follows: “10 μg/tumour and 15 μg/tumour after second virus administration”. Do the authors mean that the 10 ug dose was given after the first virus administration and the 15 ug dose after the second virus administration? Please clarify.
Round 2
Reviewer 1 Report
I thank the authors for taking the time to consider my remarks.
Although most of the minor points have been clarified, the fact that the authors themselves showed viability around 81-87% in ex vivo 4T1 tumor samples gives way to a strong / frequent nonspecific signal of different markers studied by flow cytometry. We cannot reasonably analyze these data without a viability marker, moreover when it comes to measuring population frequencies of the order of one percent, regardless of the results of preliminary experiments.
Indeed, the fact that non-viable cells, which tend to give a positive but non-specific signal, puts the conclusions of these analyzes into play dangerously.
Author Response
Please, find in attachment the revision letter.

Round 3
Reviewer 1 Report
.